# Impact of West Antarctic Ice Shelf melting on the Southern Ocean Hydrography

Yoshihiro Nakayama[1,2], Ralph Timmermann[2], and Hartmut H. Hellmer[2]

[1]Institute of Low Temperature Science, Hokkaido University, Japan
[2]Alfred Wegener Institute, Bremerhaven, Germany

**Correspondence:** Yoshihiro Nakayama (Yoshihiro.Nakayama@lowtem.hokudai.ac.jp)

**Abstract.** Previous studies show accelerations of West Antarctic glaciers, implying that basal melt rates of these glaciers were previously small and increased in the middle of the 20th century. This enhanced melting is a likely source of the observed Ross Sea (RS) freshening, but its long-term impact on the Southern Ocean hydrography has not been well investigated. Here, we conduct coupled sea-ice/ice-shelf/ocean simulations with different levels of ice shelf melting from West Antarctic glaciers. Freshening of RS shelf and bottom water is simulated with enhanced West Antarctic ice shelf melting, while no significant changes in shelf water properties are simulated when West Antarctic ice shelf melting is small. We further show that the freshening caused by glacial meltwater from ice shelves in the Amundsen and Bellingshausen seas can propagate further downstream along the East Antarctic coast into the Weddell Sea. The freshening signal propagates onto the RS continental shelf within a year of model simulation, while it takes roughly 5-10 years and 10-15 years to propagate into the region off Cape Darnley and into the Weddell Sea, respectively. This advection of freshening modulates the shelf water properties and possibly impacts the production of Antarctic Bottom Water if the enhanced melting of West Antarctic ice shelves continues for a longer period.

## 1 Introduction

Ice shelves in the Amundsen Sea (AS) and Bellingshausen Sea (BS) are melting and thinning rapidly, shown by satellite-based estimates of the last ~20 years (Depoorter et al., 2013; Rignot et al., 2013; Paolo et al., 2015), contributing significantly to ongoing ocean freshening and sea level rise through a high discharge of grounded ice (Shepherd et al., 2012; Rignot et al., 2013). The main cause for high basal melt rates is the relatively warm Circumpolar Deep Water (CDW, about 0.5–1.5 °C, located below ~300–500 m depth), which flows via submarine glacial troughs from the continental shelf break into the ice shelf cavities (e.g. Jacobs et al., 1996; Nakayama et al., 2013; Dutrieux et al., 2014; Webber et al., 2017; Jenkins et al., 2018). Against the backdrop of increased basal melt giving evidence of a sustained increase of ice discharge for most glaciers in the eastern AS since 1973 (e.g. Ferrigno et al., 1993; Lucchita and Rosanova, 1997; Rignot, 1998; Mouginot et al., 2014), there have been few studies implying that basal melt rates of these glaciers were previously small and started increasing from the middle of the 20th century (Hillenbrand et al., 2017; Smith et al., 2017).

In the Ross Sea (RS), shelf water is freshening, leading to a change in the Antarctic Bottom Water (AABW) properties (Jacobs et al., 2002; Jacobs and Giulivi, 2010). Since the salinity decrease leads to a change in AABW characteristics formed in the RS (Jacobs et al., 2002; Aoki et al., 2005; Rintoul, 2007; Jacobs and Giulivi, 2010) and may influence the global thermohaline circulation, understanding the possible link between the melting of West Antarctic ice shelves and RS freshening is important for assessing long-term changes in the Southern Ocean.

Nakayama et al. (2014a) showed the spreading pathways of glacial meltwater from ice shelves in the AS and BS, which may end up on the RS continental shelf. However, due to the difficulties in a realistic representation of Southern Ocean hydrography as well as basal melt rates, their global model simulations were limited to 10 years. Using circum-Antarctic or global domains, a few studies (Kusahara and Hasumi, 2014; Dinniman et al., 2016) also showed pathways of glacial meltwater using passive tracers, confirming that glacial meltwater from the AS and BS ice shelves flows westwards onto the RS continental shelf. Kusahara and Hasumi (2013) also performed meltwater tracer experiments in idealized warming climates, showing that increased basal meltwater from the AS and BS causes the bottom water freshening in the RS. However, the impact of the glacial melt from ice shelves in the AS and BS on the Antarctic coastal ocean has not been investigated. Recently, many ocean simulations have been designed for studying oceanographic conditions in the Amundsen Sea, but they employ regional models with more of a focus on CDW intrusions onto the Amundsen Sea continental shelf (e.g. Thoma et al., 2008; Schodlok et al., 2012; Assmann et al., 2013; St-Laurent et al., 2015; Kimura et al., 2017; Nakayama et al., 2017, 2018; Webber et al., 2019). Using global models, the impact of enhanced ice shelf melting on the global deep water circulation has been investigated (Fogwill et al., 2015; Golledge et al., 2019; Lago and England, 2019), but their model resolutions are coarse and the impact of glacial meltwater propagation in the Antarctic marginal seas has not been investigated.

After the development of the global Finite-Element Sea-ice/ice-shelf/Ocean Model (FESOM; Timmermann et al. (2012); Nakayama et al. (2014a)) including model grid refinement in the Antarctic Peninsula region and adjustment of sea ice model parameters, we are now able to carry out longer integration of our simulation with more realistic hydrographic representations of the Antarctic coastal regions. In this study, we conduct 32-year simulations to analyze the impact of glacial meltwater on the Southern Ocean. We also conduct sensitivity experiments with different ice shelf melt rates in the AS and BS.

## 2   Model

Here, we investigate ocean states using FESOM (Timmermann et al., 2012; Nakayama et al., 2014a), which includes dynamic/thermodynamic sea-ice (Timmermann et al., 2009) and thermodynamic ice shelf (Timmermann et al., 2012) capabilities. Ice shelf draft, cavity geometry, and global ocean bathymetry are derived from the RTopo-1 dataset (Timmermann et al., 2010). We use a tetrahedral mesh with a horizontal spacing of ∼100 km along non-Antarctic coasts, refined to ∼20 km along the Antarctic coast, 10-20 km under the large ice shelves in the RS and Weddell Sea (WS), and ∼5 km in the central AS and BS (Fig. 1). We apply a hybrid vertical coordinate system with 46 layers and a z-level discretization in the mid- and low-latitude ocean basins. The top 21 layers along the Antarctic coast are terrain-following (sigma coordinate) for depths shallower than 650 m. In the z-coordinate region, bottom nodes are allowed to deviate from their nominal layer depth to allow for a correct

representation of bottom topography, similar to the shaved-cell approach in finite-difference models. A Gaussian function with a width depending on the model's horizontal resolution is applied to smooth ice shelf draft and sea-floor topography in the sigma-coordinate region. Ocean bathymetry south of 55°S of the global model is shown in Fig. 1. Unlike the previous study (Nakayama et al., 2014a), no restoring is applied to any region of our model domain. The model parameters used in this study are summarized in Table 1. We assume a steady state for ice shelf thickness and cavity geometry and compute the ice shelf basal mass loss using the three-equation approach (Hellmer and Olbers, 1989) with the parametrization suggested in Holland and Jenkins (1999).

We carried out four simulations for 32 years using the ERA-Interim reanalysis product (1979-2010) (Dee et al., 2011) by changing the heat and salt transfer coefficients at the interface between ocean and those ice shelves fringing the AS and BS. For the LMELT case, these coefficients are calculated following Holland and Jenkins (1999) with the drag coefficient at the ice shelf base set to 0.0025. The coefficients are set to values three-times larger for the present (PRS) case, as the simulated ice shelf melt rates are close to present observations (Table 2). We regard the LMELT case as our reference simulation. We conduct a model spin-up of 10 years (1979-1988) using the LMELT set up. We note that ice-shelf basal mass loss for most ice shelves stabilizes within the first 5 years of model integration (Timmermann et al., 2012). The PRS case represents the transient response of the ocean to a step change of AS and BS ice shelf melting. We further conduct two other sensitivity experiments (MMELT and HMELT), which are discussed in section 3.2. To track the basal meltwater, we use a virtual passive tracer, which is released at the same rate as the glacial melt only from ice shelves in the AS and BS.

## 3   Results

### 3.1   Model Evaluation

Both LMELT and PRS produce many features of ocean circulation, water mass properties, and sea-ice distribution in good agreement with observations. The integrated transport of the Ross Gyre is $\sim$30 Sv (1Sv=$10^6$ m$^3$ s$^{-1}$) and the Antarctic Circumpolar Current (ACC) carries $\sim$160 Sv through Drake Passage. Based on oceanographic observations, the estimates of ACC transports through Drake Passage are 145.0 $\pm$8.8 and 137.9$\pm$10.5 Sv for two different vertical sections (Renault et al., 2011). Using the ocean state estimate, Mazloff et al. (2010) estimated the transport of the ACC to be 153$\pm$5.0 Sv. The simulated austral winter (September) sea-ice extent is similar to observations, slightly overestimated by 0.2 million km$^2$ (Cavalieri et al., 2006; Spreen et al., 2008), while the austral summer (March) sea-ice extent is underestimated by 1.2 million km$^2$ (Fig. S1). The variability of the modeled sea ice extent is similar to those in the observed data (Fig. S1c). The bottom temperature on the continental shelf is mostly close to the freezing point except for regions with CDW intrusions onto the continental shelves (Fig. 2). The bottom salinity exhibits local maxima towards the western WS and RS, and a zonal-shelf gradient with higher salinity at the eastern side in the AS and BS (Fig. 2). These features are present in both, the observations (Figs. 1A-B in Schmidtko et al. (2014) and Fig. 2 in Jenkins et al. (2016)) and the model results. Despite the general hydrographic features being well reproduced, the on-shelf CDW temperatures are underestimated in the AS and the BS by roughly 0.5 °C and the observed salinity gradients in the AS and BS are more strongly pronounced than in the model results.

As a result of the different heat and salt transfer coefficients, the total ice shelf basal mass losses are 192 Gt yr$^{-1}$ and 336 Gt yr$^{-1}$ in the AS, and 155 Gt yr$^{-1}$ and 260 Gt yr$^{-1}$ in the BS for LMELT and PRS, respectively (Table 2). For the AS and BS, the PRS loss rates are slightly lower than satellite-based estimates between 2003-2009 (Table 2; Depoorter et al. (2013); Rignot et al. (2013)). The total LMELT loss rate of all AS and BS ice shelves is 347 Gt yr$^{-1}$, which is $\sim$110 Gt yr$^{-1}$ smaller than the steady state melt rates (assuming zero thickening) of 461 Gt yr$^{-1}$ estimated based on the 2006-2007 ice shelf configurations (Supplementary Table in Rignot et al. (2013)). However, LMELT may better represent the melt rates in the middle of the last century, considering the fact that ice shelf cavity geometry should have evolved significantly since then (Jenkins et al., 2010a; Smith et al., 2017) and West Antarctic glaciers should have flowed much slower at the time (Mouginot et al., 2014). Time series of integrated basal melt flux of all ice shelves in the AS and BS show some variability for both LMELT and PRS. For LMELT, the melt flux decreases from 1979 to 2000, takes the minimum value of $\sim$240 Gt yr$^{-1}$ in 2001, and increases towards the end of the simulation (Fig. 3a). This evolves similarly to what was obtained in the PRS case. Thus, the total basal melt flux difference between LMELT and PRS does not exhibit large temporal variability and remains at a value of 250 Gt yr$^{-1}$ (Fig. 3b).

For the LMELT case, the impact of freshening is small as no significant changes in shelf water properties occur during the model integration, e.g., RS shelf salinity at the bottom remains stable (Fig. 4). Comparing PRS and LMELT, near-bottom CDW potential temperature and salinity in the AS and BS are higher by $\sim$0.3 °C and $\sim$0.02 in PRS than in LMELT (Figs. S2 and 4), respectively, and salinity is lower mostly elsewhere for model year 32 (Fig. S2). Both LMELT and PRS show glacial meltwater spreading downstream onto the RS continental shelf and then further along the East Antarctic coast as well as eastward to the northwestern WS within the ACC (Fig. 5; Nakayama et al. (2014a); Dinniman et al. (2016)). In response to enhanced ice shelf melting, PRS shows glacial meltwater spreading further downstream (Fig. 5b). The simulated bottom salinity difference between LMELT and PRS shows a freshening along the western AS coast (year 5, Fig. 4c), which spreads further onto the RS continental shelf (year 10-32, Fig. 4). This freshening extends down to the bottom of the RS as a result of the formation and descent of dense shelf water. We note that the RS is the only location where a large amount of glacial melt from the AS and BS reaches the deep ocean (Fig. 6). For PRS, 19% and 36% of the total glacial meltwater tracer from ice shelves in the AS and BS descend to depths of 700-1600m and 1600m-bottom, respectively, most of which are found in the deep RS after 32 years of simulation (Fig. 6).

We now compare the PRS results with recent observations. Fifty years of observations of RS dense shelf water show a salinity decline of 0.03 g kg$^{-1}$ per decade (Jacobs et al., 2002; Jacobs and Giulivi, 2010). Warming and freshening of Ross Sea Bottom Water (Purkey and Johnson, 2013) extend further westward off the Adélie Land (Aoki et al., 2005; Rintoul, 2007) and Ross Sea Bottom Water experiences a $\sim$0.1°C warming as well as a $\sim$0.01 g kg$^{-1}$ freshening between 1992-2011 at 180°E along the S04P section (Purkey and Johnson, 2013). Despite us simulating the response of the Southern Ocean to an instantaneous jump in meltwater production in the AS and BS, these features are reproduced in PRS, as the RS dense shelf water freshens by $\sim$0.045 g kg$^{-1}$ over 20 years (Fig. 4i, Table 3). This dense shelf water descends to the deep ocean causing a simulated Ross Sea Bottom Water warming and freshening of $\sim$0.02°C and $\sim$0.005 g kg$^{-1}$, respectively, over 32 years (Fig.

S3, black arrow). This Ross Sea Bottom Water warming occurs in parallel to the warming of the Antarctic Slope Current by 0.1-0.2°C (Fig. S3; ∼1000-2000 m depth).

## 3.2 Spreading of glacial meltwater from West Antarctic ice shelves

We conduct two additional sensitivity experiments and investigate the impact of enhanced ice shelf melting in the AS and BS focusing on both 200 meters depth and the seafloor. Medium and high rates of ice shelf melting (referred to MMELT and HMELT, respectively) are introduced with heat and salt transfer coefficients being set to 2-times and 30-times larger values, respectively. The total ice shelf basal mass losses are 280 Gt $yr^{-1}$ and 592 Gt $yr^{-1}$ in the AS, and 218 Gt $yr^{-1}$ and 445 Gt $yr^{-1}$ in the BS for MMELT and HMELT, respectively (Table 2).

We subtract the LMELT results from MMELT, PRS, and HMELT and use the last-2-year temporally averaged fields to investigate the impact of enhanced ice shelf melting. We calculate spatial averages for the regions indicated in Fig. 1 but using regions shallower than 1000 m and deeper than 2500 m for on-shelf 200-m and bottom spatially averaged salinity, respectively (Table 3). For MMELT-LMELT, the salinity decrease is confined mostly to the AS, BS, and RS continental shelves with a freshening of 0.025 g $kg^{-1}$ and 0.0030 g $kg^{-1}$ for the RS continental shelf and the deep RS, respectively (Table 3). Freshening in other regions is small at 200-m depth amounting to 0.0038 g $kg^{-1}$ and 0.0003 g $kg^{-1}$ for the continental shelf off Cape Darnley (CD) and the WS continental shelf, respectively (Table 3). For the PRS-LMELT case, the freshwater signal extends along the East Antarctic coast to the WS with values of 0.045 g $kg^{-1}$, 0.0048 g $kg^{-1}$, 0.0078 g $kg^{-1}$, and 0.0035 g $kg^{-1}$ for the RS shelf, deep RS, off CD, and WS shelf regions, respectively (Fig. 7, Table 3). For the HMELT-LMELT case, the spatial freshening pattern remains similar to the PRS case amounting to 0.14 g $kg^{-1}$, 0.0015 g $kg^{-1}$, 0.035 g $kg^{-1}$, and 0.016 g $kg^{-1}$ for the RS shelf, deep RS, off CD, and WS shelf regions, respectively (Fig. 7, Table 3).

Our experiments clearly show the timescales for the freshening signal to reach other regions around the Antarctic continent. For all cases, the freshening signal propagates onto the RS continental shelf within a year of model simulation (Fig. 8). It takes an additional five years for this freshening signal to become visible in the deeper part of the RS (Fig. 8). Another branch of the freshening signal further propagates near the surface along the East Antarctic coast taking roughly 5-10 years and 10-15 years to propagate into the region off CD and into the WS, respectively. Since the salinity decrease continues even after model year 32 in these regions (Figs. 8 c and d), it seems to take a long time (over 32 years) for the Southern Ocean to adjust to the new state of enhanced ice shelf melting in the AS and BS.

Previous studies showed that sea ice surrounding Antarctica expanded before 2016 and this expansion has been attributed to atmosphere changes (e.g., Turner et al. (2009)) as well as accelerated basal melting of Antarctic ice shelves (Bintanja et al., 2013). Follow-up studies, however, demonstrated that the amount of freshwater required for such water column stabilization to expand sea ice extent would require an order of magnitude higher ice shelf melt rates for the duration of the observations (Swart and Fyfe, 2013; Pauling et al., 2016). Importantly we note that the simulated impact of glacial meltwater spread on the sea ice is small. For example, the austral winter (September) sea ice extent difference between LMELT and HMELT is less than 0.1 million $km^2$ for the last year of our model simulations.

## 4 Discussion

### 4.1 Experiment design and its application

Some studies suggest that strong El Niño-Southern Oscillation in the 1940s induced anomalous on-shore CDW intrusions and triggered simultaneous groundline retreats of West Antarctic ice shelves (Steig et al., 2013; Jenkins et al., 2016; Hillenbrand et al., 2017). This likely enlarged ice shelf cavities, created stable sub-ice shelf circulations, and sustained high ice shelf melt rates in the AS until the present day. We set out to investigate the impact of enhanced ice shelf melting in the AS and BS on the Southern Ocean hydrography. However, it is difficult to design different ice shelf cavity geometry or grounding line locations to force our simulations with different levels of ice shelf melt rates. Previous studies adjusted their turbulent heat and salt exchange coefficients for the following reasons: (1) the model simulation has a bias in the water mass characteristics caused by processes not implemented or resolved in the model, e.g., ocean mixing and tidal forcing; (2) the basal melting parameterization includes uncertainty, e.g., treatment of mixed layer at the ice-ocean interface, bottom drag coefficient, etc. (Jenkins et al., 2010b); (3) the basal melt rate is sensitive to not well-known ice shelf cavity geometry including channels, grounding line location, and ocean mixing induced by a steep ice base near grounding lines. Thus, in this study, we adjust turbulent heat and salt exchange coefficients and force our model with a step-wise increase of ice shelf melting in the AS and BS. Although it may be oversimplified to assume a step-wise increase in basal melting, we emphasize that the focus of our model study is on the spreading of the AS and BS glacial melt and its impact on the downstream hydrography, which shows some similarities with observations.

We note that simulated ice shelf melt rates of the AS and BS regions in LMELT are lower than present-day estimates, although we use cavity geometry from present-day configurations. This is likely caused by the fact that (1) a horizontal resolutions of ~5 km and smoothed ice shelf drafts do not allow good representations of ice shelf cavities especially steep slopes near grounding lines, where ice shelf melting peaks with strong vertical velocity and mixing (e.g., Nakayama et al. (2019); Shean et al. (2019)) and (2) the FESOM simulation has a bias in water mass characteristics and consequently simulated on-shelf CDW intrusions are weaker than those observed (Nakayama et al., 2014b).

### 4.2 Response to the enhanced ice shelf melting

We show that the glacial meltwater spreads differently depending on the magnitudes of ice shelves melting in the AS and BS. For the RS shelf and deep RS, magnitudes of freshening linearly increase as ice shelf melting in the AS and BS is enhanced (total melt rate difference in Table 3). For example, for MMELT, PRS, and HMELT basal melting increases by $150\,\mathrm{Gt\,yr^{-1}}$, $250\,\mathrm{Gt\,yr^{-1}}$, and $695\,\mathrm{Gt\,yr^{-1}}$ and simulated RS shelf region freshens by $0.025\,\mathrm{g\,kg^{-1}}$, $0.045\,\mathrm{g\,kg^{-1}}$, and $0.14\,\mathrm{g\,kg^{-1}}$, respectively. Similarly, a linear relation can be found for the deep RS (Table 3). On-shelf freshening in the RS shelif extends along the East Antarctic coast and into the WS, similar to the idea presented by Beckmann and Timmermann (2001). Freshening off the CD and in the WS, however, responds differently. Weak (or almost no) freshening is simulated for MMELT and PRS, and enhanced freshening is only simulated in the HMELT case (Table. 3). This implies that the large-scale freshening becomes significant only when ice loss in the AS and BS region is high. Due to much stronger seasonal and interannual variability in

the near-surface layers and a relatively sluggish response over the 15 to 20 model years (Figs. 8c and d), the effect of enhanced ice shelf melting cannot clearly be detected in the existing observations for these regions. However, considering the magnitude of the observed salinity decrease by now, circum-Antarctic freshening is likely to occur.

Such response of Antarctic coastal regions can not be explained by an increase of ice shelf melt rates alone. Instead we consider the freshening to be a result of the strengthening of the westward flowing coastal current due to an increased density gradient (caused by shelf water salinity) across the Antarctic Slope Front (Nakayama et al. (2014a)). For the HMELT case, the development of strong density gradients is simulated in the BS, AS, and RS, along the East Antarctic coast, and in the WS (Fig. 9c). However for the other cases, strong density gradients are simulated only in the BS, AS, and RS (Figs. 9a-b) and in a limited region along the East Antarctic coast (Fig. 9).

We also note that enhanced ice shelf melting modifies the properties of RS shelf water (Figs. 7 and 8), possibly with consequences for the global thermohaline circulation. For example, Fogwill et al. (2015) show that enhanced ice shelf melting in the AS region may lead to a significant decrease in the rate of AABW formation. In HMELT, however, properties of Ross Sea surface and bottom water towards the end of the simulation converge with a freshening of $\sim$0.12 and $\sim$0.015, respectively, rather than showing a rapid drawdown (Fig. 8). Further studies with longer model integration times are required to investigate the impact of enhanced ice shelf melting on deep water properties.

## 5 Conclusions

In this study, we conduct four 32-year simulations with different levels of ice shelf melting in the AS and BS to investigate the impact of glacial meltwater on the Antarctic continental shelf hydrography. The total ice shelf mass losses from the AS and BS are 346 Gt yr$^{-1}$, 497 Gt yr$^{-1}$, 596 Gt yr$^{-1}$, and 1036 Gt yr$^{-1}$ for LMELT, MMELT, PRS, and HMELT, respectively (Tables 1 and 2). We show that the LMELT result represents a quasi-steady state without significant change in RS shelf water salinity (Fig. 4), and the PRS result shows a RS continental shelf and deep ocean freshening with some similarities to recent observations (Fig. 4). In addition, we show that glacial meltwater from the AS and BS may propagate further downstream along the East Antarctic coast leading to salinity decreases off CD and in the WS in PRS, and HMELT (Figs. 7 and 8). The freshening signal propagates onto the RS continental shelf within a year of model simulation, while it takes roughly 5-10 years and 10-15 years to propagate into the region off Cape Darnley and into the Weddell Sea, respectively. This modulates the shelf water properties and may reduce the production of AABW if the enhanced melting of West Antarctic ice shelves continues for a longer period. We also show that the amount of freshening observed in the Ross Sea surface and bottom waters increases linearly as the freshwater flux from ice shelf melting in the AS and BS increases (Table 3). However, for regions further downstream, off CD and in the WS, the impact of freshening can be detected only when ice loss in the AS and BS is significant. Such response of the Antarctic coastal regions is likely related to the development of a strong cross-slope front density gradient caused by lower shelf water salinity, and thus a more vigorous westward flowing coastal current. Considering the spatial and temporal scales of AS and BS glacial meltwater spreading around Antarctica, further investigations and model

developments are required to understand the impact of West Antarctic ice shelf melting on the circum-Antarctic and global ocean.

*Code availability.* Model codes presented in this study are available in https://fesom.de.

*Data availability.* The model grid and output are provided in http://www.lowtem.hokudai.ac.jp/wwwod/nakayama/.

*Author contributions.* YN prepared the manuscript, conducted ocean simulations. RT and HH helped interpreting the results. All authors commented on the manuscript.

*Competing interests.* The authors declare no competing interests.

*Acknowledgements.* We thank Dmitry Sidorenko and Lukrecia Stulic for their numerical and technical support and Sunke Schmidtko for
sharing the observed data. Alex Gardner, Andrew Thompson, Dimitris Menemenlis, Eric Rignot, Surendra Adhikari, Stan Jacobs, and Shigeru Aoki provided helpful comments and suggestions. Simulations were carried out at the Supercomputing Division, Information Technology Center at University of Tokyo and at the North-German Supercomputing Alliance (HLRN). We acknowledge funding by the Helmholtz Climate Initiative REKLIM (Regional Climate Change), a joint research project of the Helmholtz Association of German Research Centres (HGF). This work was also supported by the fund from Grant in Aids for Scientific Research (19K23447) of the Japanese Ministry of
Education, Culture, Sports, Science and Technology. Insightful comments from two anonymous reviewers and Asay-Davis Xylar were very helpful for improving the manuscript.

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

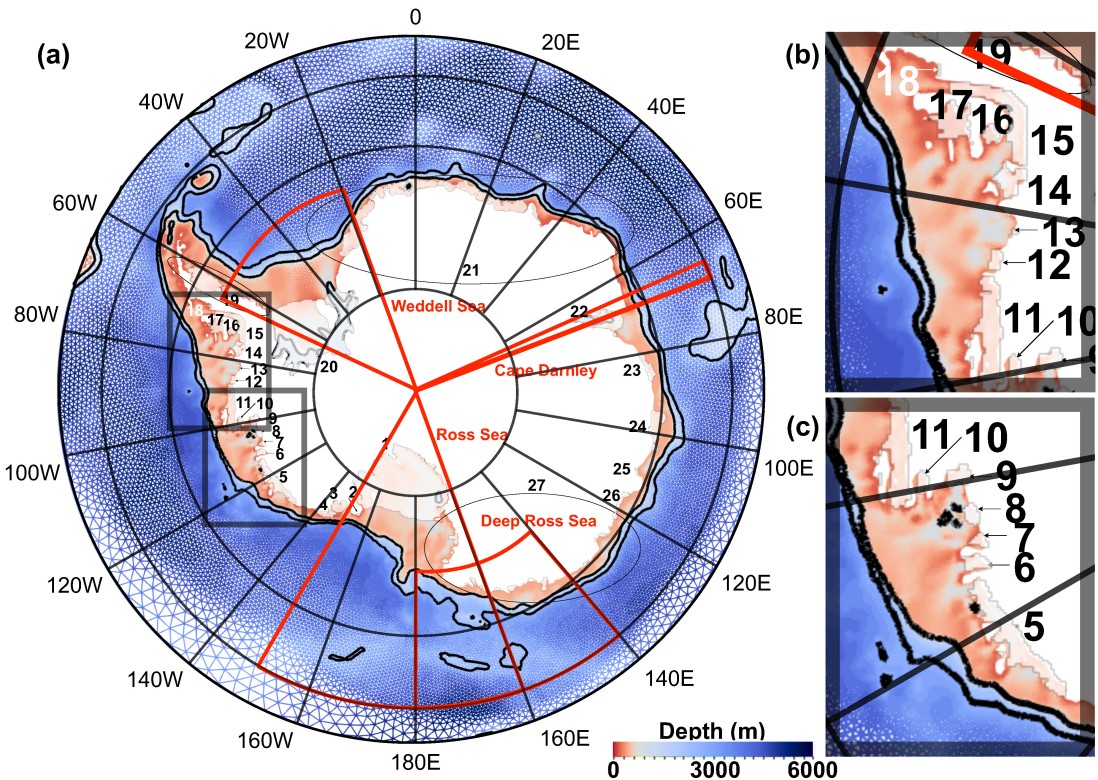

**Figure 1.** (a) Horizontal grid (triangles) and model bathymetry (color) south of 55ºS in the global model. The bathymetry contours of 1000 m and 2500 m are shown in black lines. Locations of ice shelves are indicated by numbers summarized in Table 3. Basal melt rates are integrated for several ice shelves in the WS and East Antarctica, bordered by ellipses, for model-data comparison (Table 3). The regions enclosed by red lines represent the RS, deep RS, CD, and WS, in which spatially averaged water mass characteristics are calculated (Table 4 and Fig. 8). Close-ups for the Bellingshausen and Amundsen seas enclosed by the black boxes in (a) is shown in (b) and (c), respectively.

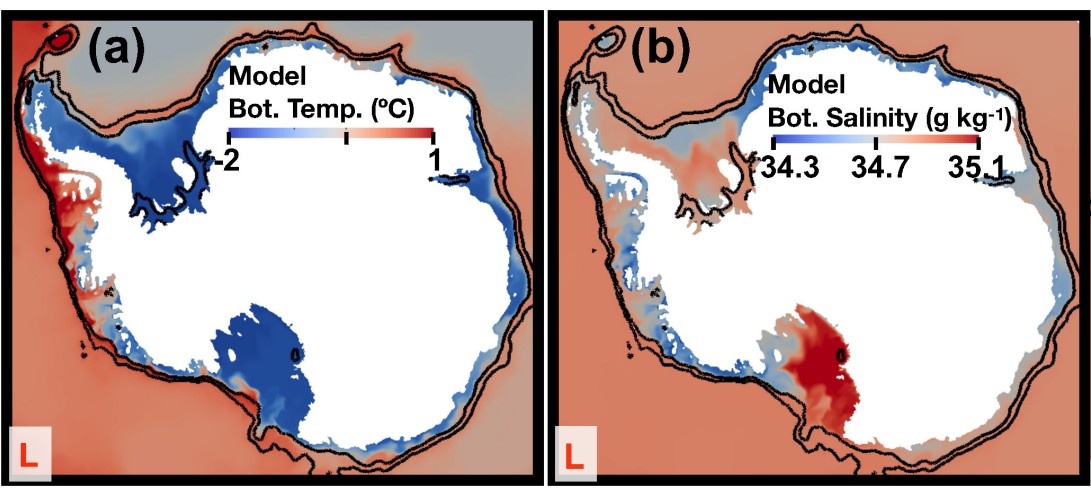

**Figure 2.** January mean bottom properties for (a) potential temperature and (b) absolute salinity for the LMELT (L) case for model year 32. The bathymetry contours of 1000 m and 2500 m are shown in black lines.

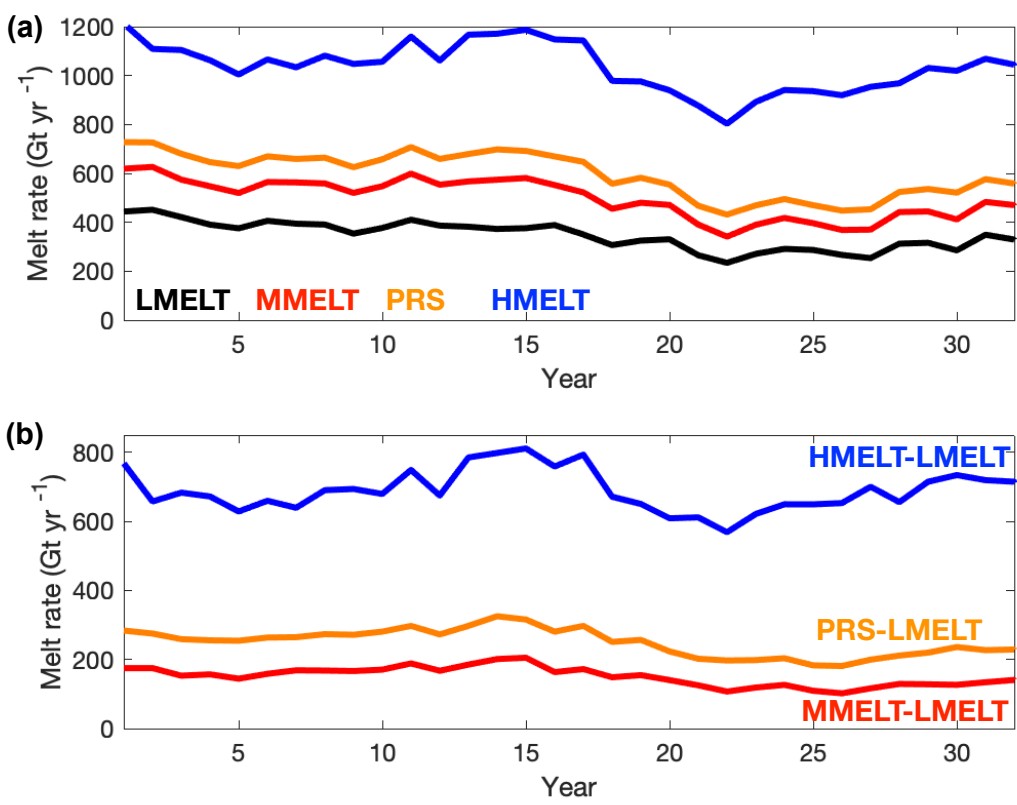

**Figure 3.** Time series of area integrated annual mean melt rates for (a) Amundsen and Bellingshausen seas for LMELT, MMELT, PRS, and HMELT. (b) Difference of integrated annual mean melt rates for AS and BS ice shelves between HMELT and LMELT (blue), PRS and LMELT (orange), and MMELT and LMELT (red).

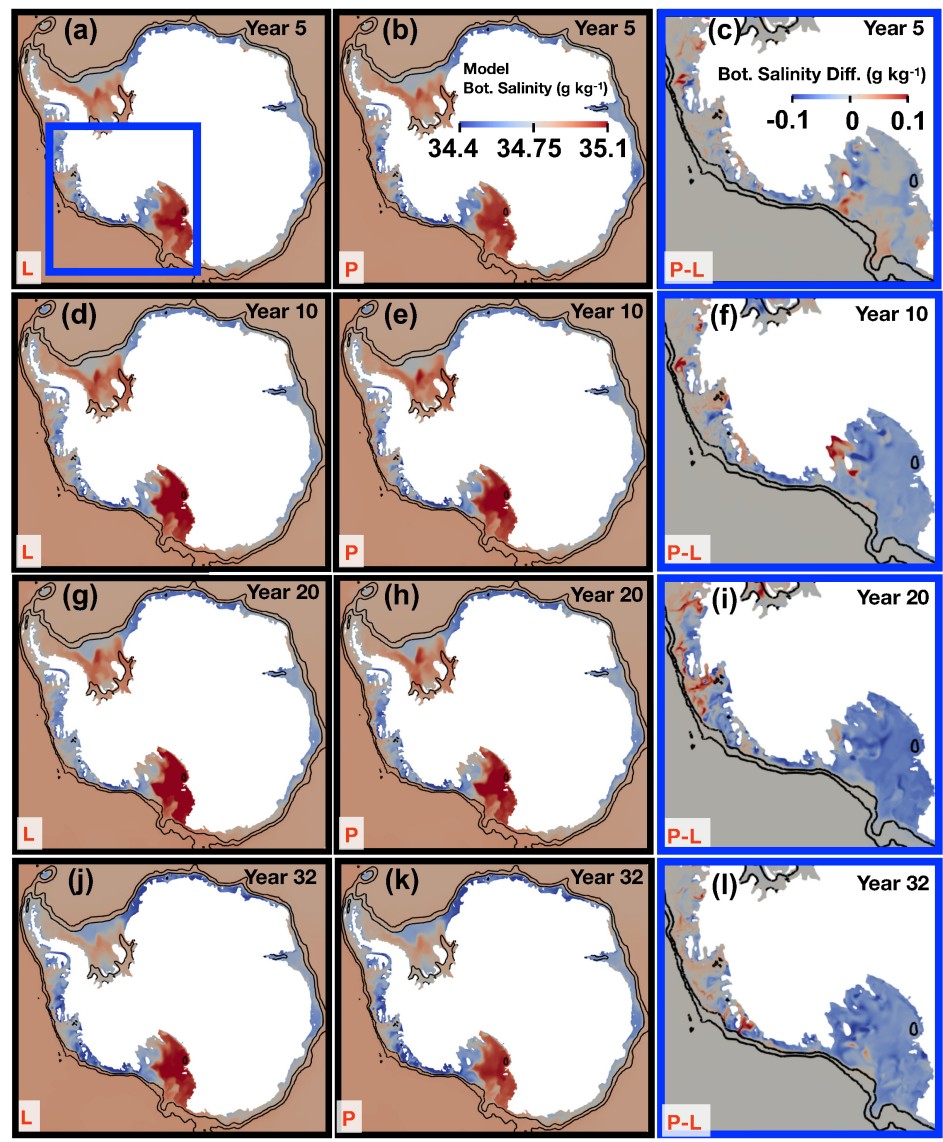

**Figure 4.** January mean bottom absolute salinity for LMELT (L) and PRS (P) and the differences (P-L) for years 5, 10, 20, and 32. The differences are shown for the region enclosed by the blue box in (a). The bathymetry contours of 1000 m and 2500 m are shown in black lines.

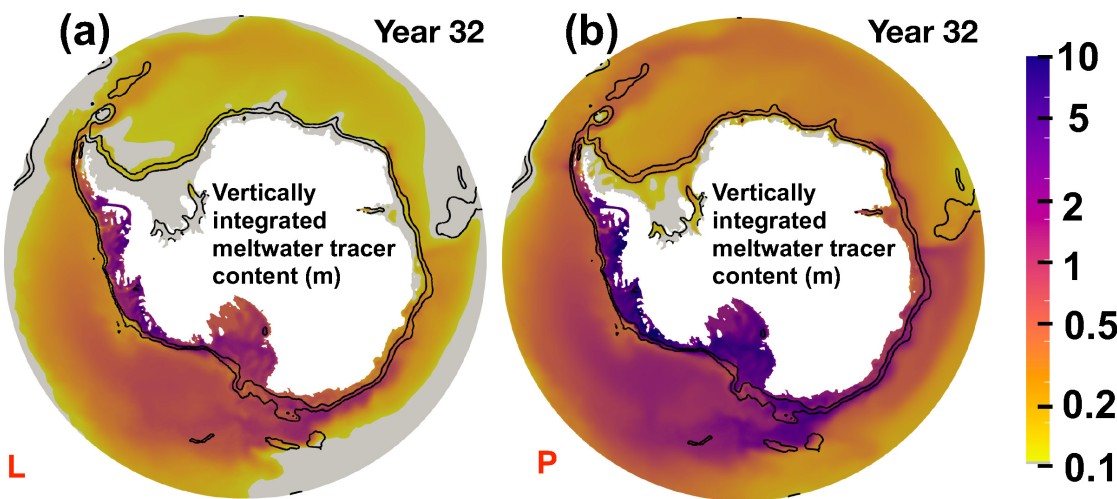

**Figure 5.** January mean vertically integrated tracer contents representing the glacial meltwater only from ice shelves in the AS and BS for (a) LMELT and (b) PRS cases for year 32. The letters P and L at the bottom left of each panel indicate PRS and LMELT, respectively. The bathymetry contours of 1000 m and 2500 m are shown in black lines. Values lower than 0.1 are indicated in gray.

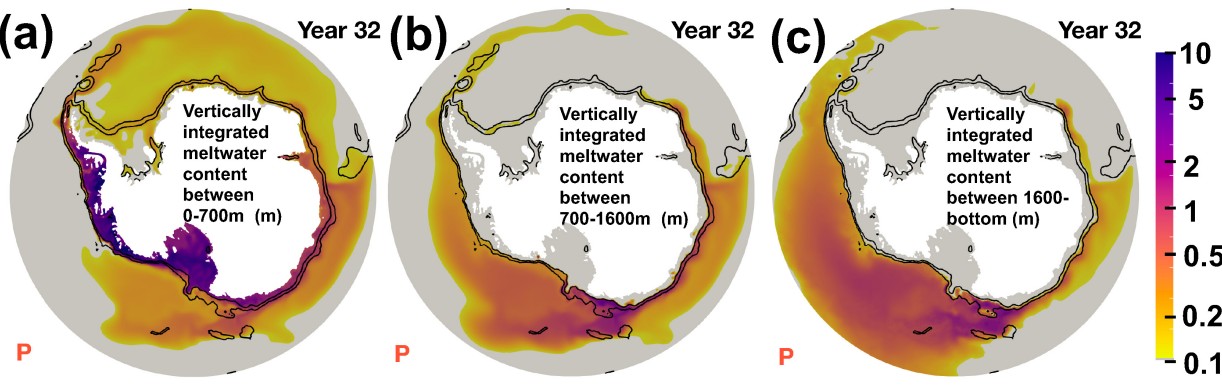

**Figure 6.** January mean vertically integrated glacial meltwater content between (a) 0-700 m, (b) 700-1600 m, and (c) 1600 m to bottom of year 32 for PRS case (P). The bathymetry contours of 1000 m and 2500 m are shown in black lines. Values lower than 0.1 are indicated with gray.

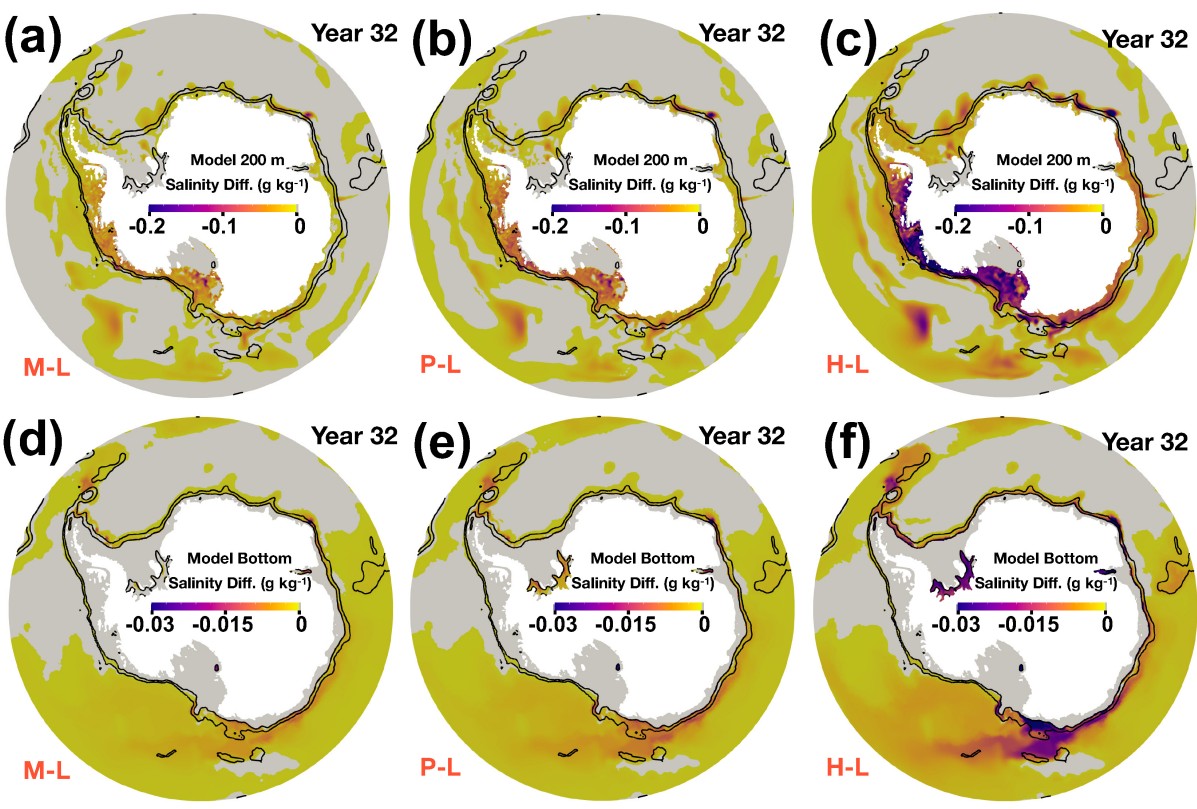

**Figure 7.** January mean absolute salinity differences MMELT-LMELT (M-L) for year 32 at (a) 200-m depth and (d) bottom. Same for PRS-LMELT (P-L), and HMELT-LMELT (H-L) shown in (b, e) and (c, f), respectively. Bottom properties are only shown for regions deeper than 1500m. The bathymetry contours of 1000 m and 2500 m are shown in black lines. Positive values are indicated in gray.

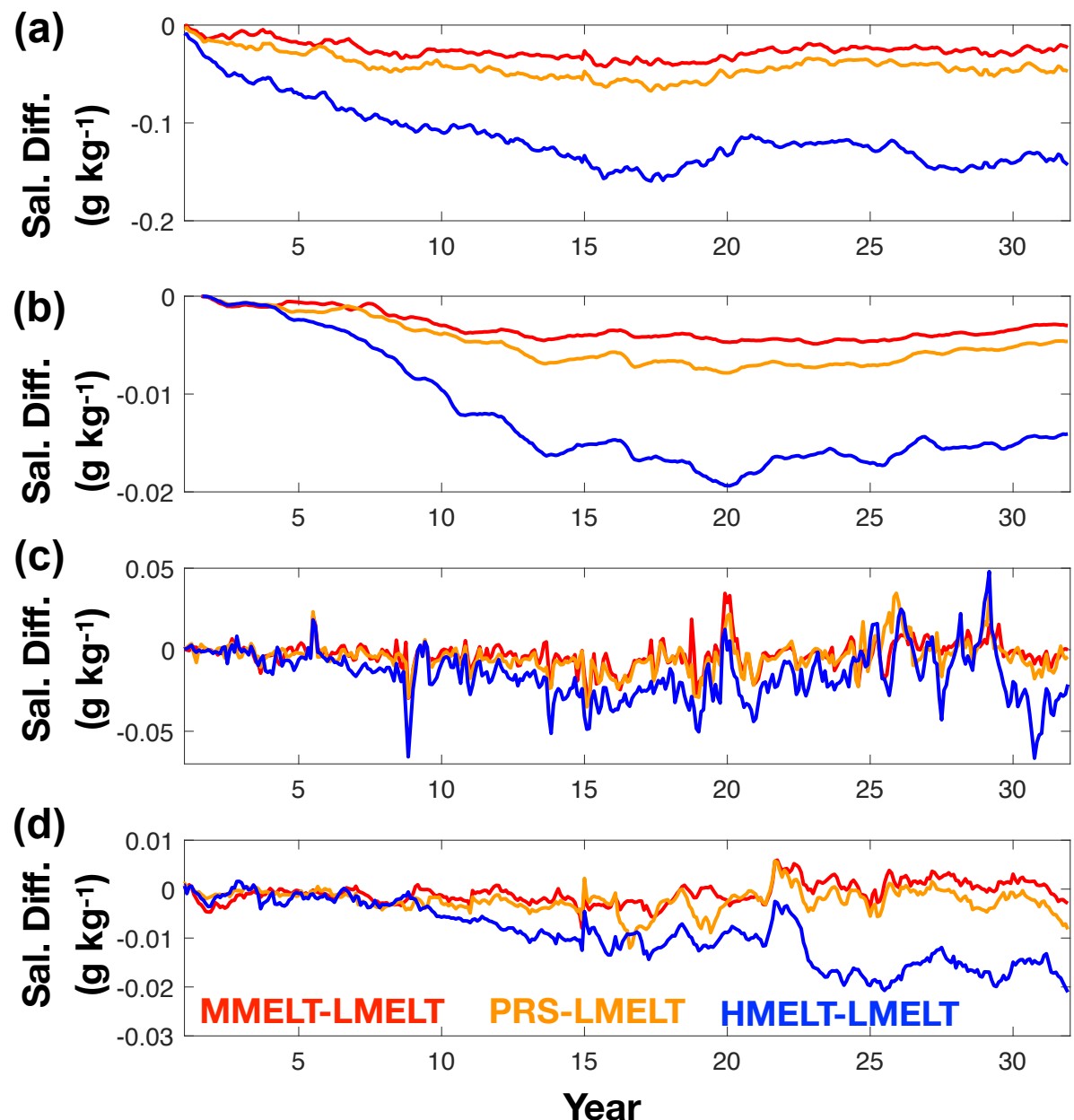

**Figure 8.** Time series of spatially averaged salinity difference over (a) RS continental shelf at 200-m depth, (b) deep RS bottom, (c) continental shelf off CD at 200-m depth, and (d) WS continental shelf at 200-m depth. Spatial averages have been calculated for the regions indicated in Fig. 1 but using regions shallower than 1000 m and deeper than 2500 m for on-shelf 200-m spatially averaged and bottom spatially averaged salinity, respectively (Table 4). LMELT fields are subtracted from HMELT (blue), PRS (orange), and MMELT (red) fields to calculate the differences.

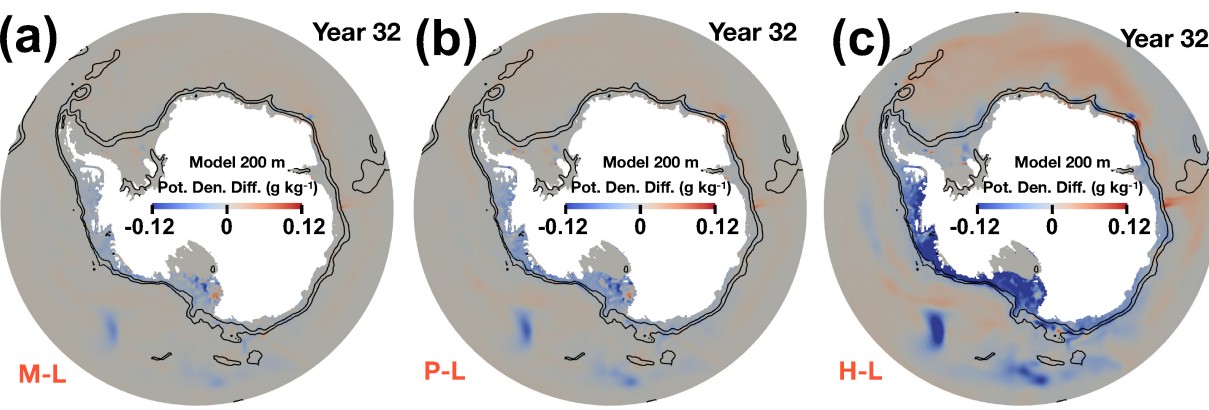

**Figure 9.** January mean potential density difference for (a) MMELT-LMELT (M-L), PRS-LMELT (P-L), and (c) HMELT-LMELT (H-L) for year 32 at 200-m depth. The bathymetry contours of 1000 m and 2500 m are shown in black lines.

**Table 1.** Model parameters used for the simulations in this study. Horizontal diffusivity and viscosity are scaled by area. For example, a grid element with an area of $5.8{\times}10^9$ m$^2$ or a 1.5° triangular grid yields a viscosity of $6.0{\times}10^4$ m$^2$ s$^{-1}$ and a diffusivity of about $1.8{\times}10^3$ m$^2$ s$^{-1}$.

| Parameter | |
|---|---|
| Horizontal diffusivity scaling factor (m$^2$ s$^{-1}$) | $1.8 \times 10^3$ |
| Background horizontal viscosity scaling factor (m$^2$ s$^{-1}$) | $6.0 \times 10^4$ |
| Scaling reference area (m$^2$) | $5.8 \times 10^9$ |
| Background vertical diffusivity (m$^2$ s$^{-1}$) | $5.0 \times 10^{-5}$ |
| Background vertical viscosity (m$^2$ s$^{-1}$) | $1.0 \times 10^{-3}$ |
| Bottom drag coefficient | $2.5 \times 10^{-3}$ |
| Air/sea ice drag coefficient | $2.5 \times 10^{-3}$ |
| Sea ice/ocean drag coefficient | $5.0 \times 10^{-3}$ |
| Sea ice salt concentration (g kg$^{-1}$) | 5.0 |
| Drag coefficient at ice shelf base | $2.5 \times 10^{-3}$ |
| Lead closing (m) | 0.1 |
| Ice strength (N m-2) | $1.5 \times 10^4$ |
| Sea ice dry albedo | 0.75 |
| Sea ice wet albedo | 0.68 |
| Snow dry albedo | 0.85 |
| Snow wet albedo | 0.77 |

**Table 2.** Antarctic ice shelf basal mass loss from LMELT, MMELT, PRS and HMELT and satellite-based estimates [Depoorter et al., 2013, Rignot et al., 2013]. Ice shelf locations are indicated by numbers in Fig. 1. For some ice shelf regions in the Weddell Sea and East Antarctica ((19)-(22) and (24)-(27), respectively), basal melt rates are accumulated for several ice shelves and compared to the satellite-based estimates as indicated in Fig. 1. Turbulent heat and salt exchange coefficients for ice shelves in AS and BS (bold) are increased in sensitivity experiments.

| Name | LMELT (Gt yr$^{-1}$) | MMELT (Gt yr$^{-1}$) | PRS (Gt yr$^{-1}$) | HMELT (Gt yr$^{-1}$) | Satellite-based estimates (Gt yr$^{-1}$) | References |
|---|---|---|---|---|---|---|
| (1) Ross | 110.2 | 110.5 | 110.3 | 110.7 | 14-82 | Rignot et al., 2013; Depoorter et al., 2013 |
| (2) Withrow | 0.1 | 0.1 | 0.1 | 0.1 | -0.1-0.7 | Rignot et al., 2013 |
| (3) Swinburne-Salzberger | 16.5 | 15.6 | 14.9 | 12.8 | 19-26 | Rignot et al., 2013 |
| (4) Nickerson-Land | 3.0 | 2.9 | 2.8 | 2.7 | 5-11 | Rignot et al., 2013 |
| **(5) Getz** | 93.3 | 139.4 | 168.3 | 309.4 | 117-159 | Rignot et al., 2013; Depoorter et al., 2013 |
| **(6) Dotson** | 13.1 | 19.6 | 23.2 | 33.1 | 41-49 | Rignot et al., 2013 |
| **(7) Crosson** | 3.5 | 4.7 | 5.3 | 5.8 | 35-43 | Rignot et al., 2013 |
| **(8) Thwaites** | 15.2 | 22.2 | 27.0 | 48.3 | 91-105 | Rignot et al., 2013 |
| **(9) Pine Island** | 28.3 | 42.2 | 52.6 | 103.4 | 81-109 | Rignot et al., 2013; Depoorter et al., 2013 |
| **(10) Cosgrove** | 10.3 | 15.9 | 20.3 | 37.2 | 7-11 | Rignot et al., 2013 |
| **(11) Abbot** | 28.5 | 35.5 | 39.4 | 54.5 | 33-97 | Rignot et al., 2013; Depoorter et al., 2013 |
| **(12) Venable** | 2.4 | 3.6 | 4.2 | 7.3 | 17-21 | Rignot et al., 2013 |
| **(13) Ferrigno** | 0.1 | 0.1 | 0.2 | 0.7 | 3-7 | Rignot et al., 2013 |
| **(14) Stange** | 23.6 | 34.6 | 41.9 | 79,1 | 22-34 | Rignot et al., 2013 |
| **(15) George VI** | 104.4 | 147.7 | 176.8 | 298.7 | 72-160 | Rignot et al., 2013; Depoorter et al., 2013 |
| **(16) Bach** | 4.7 | 7.0 | 8.9 | 17.4 | 9-11 | Rignot et al., 2013 |
| **(17) Wilkins** | 19.3 | 25.1 | 28.3 | 41.4 | 1-35 | Rignot et al., 2013 |
| **(18) Wordie** | 0.1 | 0.3 | 0.4 | 1.7 | 4-10 | Rignot et al., 2013 |
| (19) Larsen B-G | 36.4 | 36.6 | 36.0 | 35.3 | -59-134 | Rignot et al., 2013; Depoorter et al., 2013 |
| (20) Filchner-Ronne | 108.5 | 106.9 | 107.8 | 109.1 | 10-200 | Rignot et al., 2013; Depoorter et al., 2013 |
| (21) Brunt-Downer | 101.3 | 101.2 | 101.0 | 100.1 | 40-162 | Rignot et al., 2013 |
| (22) Amery-Publication | 64.8 | 64.3 | 63.8 | 62.1 | 12-62 | Rignot et al., 2013; Depoorter et al., 2013 |
| (23) West | 14.6 | 14.7 | 14.8 | 15.0 | 17-37 | Rignot et al., 2013 |
| (24) Shackleton-Glenzer | 22.1 | 22.3 | 22.3 | 22.2 | 61-97 | Rignot et al., 2013 |
| (25) Vincennes | 1.1 | 1.1 | 1.1 | 1.0 | 3-7 | Rignot et al., 2013 |
| (26) Totten-Moscow Univ. | 9.7 | 9.6 | 9.6 | 9.7 | 83-99 | Rignot et al., 2013 |
| (27) Holmes-Drygalski | 23.4 | 22.2 | 21.3 | 18.4 | 33-72 | Rignot et al., 2013 |
| Amundsen Sea (5-11) | 192.2 | 280.0 | 336.0 | 591.7 | 405-573 | Rignot et al., 2013; Depoorter et al., 2013 |
| Bellingshausen Sea (12-18) | 154.6 | 218.1 | 260.3 | 444.7 | 128-278 | Rignot et al., 2013; Depoorter et al., 2013 |
| Antarctic total | 837.4 | 976.3 | 1068.0 | 1480.1 | 1263-1737 | Rignot et al., 2013; Depoorter et al., 2013 |

**Table 3.** Differences of mean Antarctic ice shelf melt rate and spatially averaged salinity for different regions for the last 2 years of model simulations. The LMELT field is subtracted from HMELT, PRS, and MMELT fields to calculate the differences. We calculate spatial averages for the regions indicated in Fig. 1 but using regions shallower than 1000 m and regions deeper than 2500 m for on-shelf 200-m spatially averaged and bottom spatially averaged salinity, respectively.

| | HMELT-LMELT | PRS-LMELT | MMELT-LMELT |
|---|---|---|---|
| Total melt rate difference (Gt yr$^{-1}$) | 643 | 231 | 138 |
| RS continental shelf salinity difference at 200-m depth (g kg$^{-1}$) | -0.14 | -0.045 | -0.025 |
| Deep RS salinity difference at bottom (g kg$^{-1}$) | -0.015 | -0.0048 | -0.0030 |
| Continental shelf region off Cape Darnley salinity difference at 200-m depth (g kg$^{-1}$) | -0.035 | -0.0078 | -0.0038 |
| Weddell Sea continental shelf salinity difference at 200-m depth (g kg$^{-1}$) | -0.016 | -0.0035 | -0.0003 |