# Peer review of "Impact of West Antarctic Ice Shelf melting on the Southern Ocean Hydrography"

_The Cryosphere, 2019_

## Referee Comment (RC1) · Anonymous Referee #1 · 30 Dec 2019

General comments

In this study the authors use the finite element ocean/sea ice/ice shelf FESOM model to study the impact of increased basal melting of the ice shelves in the Amundsen (AS) and Bellingshausen (BS) Seas on the hydrography of the entire Antarctic continental shelf and the condition of Antarctic Bottom Water (AABW) just off the continental shelf. This is done through examining four 32 year long simulations where the ice shelf basal melt rates are increased between simulations by modifying the transfer coefficients between the ice and the water underneath. The freshening signal not only propagates onto the Ross Sea continental shelf, but within the time frame of these simulations also makes its way around almost the entire continent onto the Weddell Sea continental shelf. The increased melt scenarios also impact the AABW off the Ross Sea and have

slight impacts on AABW elsewhere.

I thought the paper was generally clear and well written. The impacts of increasing ice shelf basal melt in the AS and BS on Antarctic continental shelf waters and AABW are an important problem and, in my opinion, well worth the attention of the Cryosphere. FESOM, with its high resolution on the Antarctic continental shelf (including under the ice shelf cavities) and slope, explicit ice shelves, and global domain (so no worries about lateral boundary conditions) is a fantastic tool to study this question.

My only negative general comment is relatively minor, but I do think there are some warnings about the applicability of these results that should be included. The LMELT results are using what the authors think the heat and salt transfer coefficients should be, but this results in low basal melting compared to observations. No mention is made of why they think the melting is low compared to current day conditions: Is this because of unknown ice/ocean interaction physics or is there a problem with the representation of water masses on the continental shelf? If it's an issue with the water masses, does this influence the rate at which meltwater advects (due to lateral density gradients) in the coastal current over the continental shelf? Also, the HMELT increased melting is not due to changes in the AS/BS shelf conditions, as they presumably are in the real world since the mid-20th century, but rather numerical manipulation of the ice/ocean transfer coefficients. Does this have an impact on the results?

I have some other specific comments and suggestions below, but most of these are very minor and should be easily dealt with by the authors.

Specific comments

Abstract, line 3: The abstract states that the long term impact of enhanced melting of the Amundsen Sea ice shelves "on the Southern Ocean hydrography has not been well investigated". However, there have been several studies of this (e.g. Fogwill et al., 2015; Golledge et al., 2019; Lago and England, 2019), just not with models setup as nicely as the FESOM model used here (i.e. explicit ice shelves and high resolution

around Antarctica). I think it would be helpful to mention some of the other studies in the Introduction, but also include mention of why the model used here is better suited for examining this question.

Abstract, line 7 and line 155: See comments below about the propagation of the meltwater, but suggest changing "propagates further" to "can propagate further".

First paragraph of model section: Even though the authors mention the ice/ocean heat/salt transfer coefficients in the next paragraph, I think it would be helpful to readers not familiar with FESOM to explicitly mention in this paragraph that FESOM does simulate the melting/freezing of the base of the floating ice shelves. Should also explicitly mention that FESOM does have a dynamic sea ice model.

Lines 71-73: I think it would be helpful if the authors added a figure about the simulated sea ice extent to the Supplement.

Lines 124-126: The HMELT case shows the propagation of the freshening signal as described here, but it's often hard to see if there has been a propagation of the signal in the other cases. For example, the red and orange lines in Figure 6c do not stay below zero until ∼ year 15 and then go back above zero for a good portion of the time past year 20. In 6d, one could argue that the red and orange lines do not stay below zero until almost the end of the period. This is why I suggested the change in line 7 of the Abstract/line 155.

Line 151: I think it's a bit much to say this paper is investigating the impact of the meltwater on "the Southern Ocean hydrography". It is looking at some aspects of the hydrography (Antarctic continental shelf conditions and changes in AABW), but not at all the broad scale water masses that are involved in the Southern Ocean. Suggest changing "Southern Ocean hydrography" to something a little more focused.

Technical corrections

Line 15: To avoid confusion from some readers about ice shelf vs. grounded ice contributions to sea level rise, suggest changing "ongoing sea level rise and ocean freshening" to "ongoing ocean freshening as well as to sea level rise".

Line 21: Suggest changing "There exist a few other evidences" to "There is some evidence".

Line 35: Suggest changing "focuses" to "of a focus".

Lines 60 and 61: Are the transfer coefficients set to constants as in Hellmer and Olbers or functions of the friction velocity as in Holland and Jenkins? From other FESOM ice shelf papers, I assume they are functions of the friction velocity, but I can't tell from how it is written here.

Line 82: From Rignot et al. (2013), I get 664 Gt/yr (not 459) for their estimate of the basal melt of the combined AS and BS (numbers 5-18 in Table S3) ice shelves.

Line 83: Change "at that the time in the middle" to "in the middle".

Lines 87-88 and Figure 3: If the Figure 3 plots are mean bottom salinity, then how does this show that the salinity at 200-m depth is stable? Is "bottom" over the continental shelf in the figure defined at 200-m?

Line 89: Suggest changing "the RS continental shelf further along the east Antarctic coast and towards" to "the RS continental shelf and then further along the east Antarctic coast as well as towards".

Line 92: Typo, "Fig .3" should be "Fig. 3".

Line 101: Suggest changing "Despite underestimated" to "Despite being underestimated".

Line 143: Typo, "0.030" should be "0.0030" and "0.048" should be "0.0048" (assuming Table S4 is correct).

Line 167: Add "on" after "commented".

Figure 2: Why does the temperature scale top out at 1.0C? The Schmidtko et al. observations have the mean BS temperature > 1.0, and thus it's hard to make comparisons between the model and the observations in the AS and BS continental shelves.

Table S1: What are the units for the sea ice salt concentration and is the value here correct? Timmermann et al. (2009) has it as 5 (psu or g/kg).

Table S3: I don't understand what "16" and "17" are in the references. I assume one is Depoorter et al. and one is Rignot et al., but can't tell which is which.

---

## Referee Comment (RC2) · Anonymous Referee #2 · 16 Jan 2020

General Comments:

Many satellite and oceanographic observations revealed that Antarctic Ice sheets and the Southern Ocean have been changing in recent decades. The interaction between Antarctic ice sheets/shelves and the Southern Ocean is one of the most important topics in the climate sciences. This study investigates pathways of ice-shelf meltwater from the West Antarctic ice shelves and its role on the Southern Ocean conditions, using a series of sea-ice/ice-shelf/ocean simulations. In my reading, the results of this study will be a valuable contribution to the Antarctic sciences. I recommend publication in The Cryosphere after addressing the comments listed below.

I have two major comments.

(1) This paper used numerical experiments with different levels of basal melting (by tuning the transfer coefficients) to explain the observed changes (e.g., lines 97-104). In my reading, the results from CTRL (or CTRL-LMELT) shows a transition from the LMELT conditions and are not suitable for explaining the observed changes. The transition timescale is useful information, but the comparison of the Southern Ocean water properties between the model and observation in the present manuscript may be misleading.

(2) Although there are sentences about the impact of the meltwater on AABW formation in the remote regions (Cape Darnley and Weddell Sea) in abstract and discussion (lines 9-11 and158-159), Figure 5d-f show no pronounced change in the bottom water properties. I understand the idea, but the simulations didn't support it.

Specific comments:

(3) lines 31-34: Wrong and missing citations Kusahara et al. (2017) is a modeling study of dense shelf water, not ice-shelf meltwater. Kusahara and Hasumi (2013, JGR-Oceans) performed virtual (meltwater) tracer experiments in idealized warming climates, showing that increased basal meltwater from the Amundsen and Bellingshausen Seas causes the bottom water freshening in the Ross Sea and Australia-Antarctic Basins.

(4) lines 38-40 Please briefly explain what kind of model development allows the longer integration.

(5) The description of ice-ocean interaction is missing.

(6) I think that 10-years spin-up is short.

(7) Lines 97-104 and Fig. S2a What is the mechanism of the bottom water warming in the Ross Sea?

(8) All map figures need longitude and latitude information (at least one panel).

(9) The manuscript is not so long. I suggest merging the supplementary material into the main text to increase readability.

---

## Referee Comment (RC3) · Xylar Asay-Davis (Referee) · 18 Jan 2020

**Review of Nakayama et al. "Impact of West Antarctic Ice Shelf melting on the Southern Ocean Hydrography"**

Reviewer: Xylar Asay-Davis

I wish my name to be relayed to the authors, as I do not support the practice of anonymous review.

**General Comments:**

Main points:
- Freshwater from the Amundsen and Bellingshausen (AB) Seas is shown to reach the Ross continental shelf in XXX years, the deeper Ross Sea within XXX years, the region near the Amery Ice Shelf after XXX years and the Weddell Sea in XXX years.
- For the most significant amounts of melting, on the order of 10 times currently observed melt rates in the AB region, freshwater reaches the Weddell Sea much more quickly (~10 years into the simulation) and the amount of freshwater reaching the Weddell continental shelf is enough to reduce the salinity there by a non-negligible amount.
- In simulations with AB melt rates comparable to or less than present-day, meltwater may reach the Weddell Sea after ~30 years but its impact on salinity are difficult to distinguish from temporal (and perhaps ensemble) variability
-

I get the impression in several places in the text that the experiments were designed (and perhaps the manuscript was originally written) with LMELT as the intended control experiment, and that perhaps a later decision was made that CTRL should be the control because its melt rates are most comparable to observations. Differences are repeatedly taken with respect to LMELT, rather than CTRL and the coefficients used in LMELT are stated to be the ones commonly used in other FESOM simulations. I would urge you to consider explicitly renaming LMELT to CTRL and CTRL to something else because this would seem more consistent with the manuscript as written. Several specific comments point out in more detail where this inconsistency arises.

The manuscript seems short for TC, especially the discussion section (see below). It sometimes reads as if it were intended for a journal that requires a shorter page count. This may explain why several tables that are referenced repeatedly in the text are included in the supplement rather than the main manuscript. I am not aware of a limit on tables or figures that require putting so many tables in the supplement. In particular, values from Table S4 are repeated (in multiple places) in the text, which would presumably not be necessary if that table were in the main text.

Speaking of which, there is a lot of redundancy both within the text vs. in tables and between the results, discussion, and conclusion sections. I have pointed out where I find this redundancy in the specific comments. This redundancy comes at the expense of what could

have been a broader discussion of the results that synthesizes the findings in a somewhat more qualitative fashion and talks about their broader implications based on observed and projected changes in AB melting, impacts of freshening on both the Ross continental shelf, deeper Ross Sea and elsewhere, etc.

Colormaps are not very intuitive and are not friendly to readers with color blindness. The manuscript preparation guidelines include the following: "For maps and charts, please keep colour blindness in mind and avoid the parallel usage of green and red. For a list of colour scales that are illegible to a significant number of readers, please visit ColorBrewer 2.0." In addition to concerns about color blindness, the colormaps used in this manuscript suffer from alternate banding of bright and dark colors that make it difficult for a reader to intuitively tell higher from lower values of the field. (In the terminology of color theory, they are not perceptually uniform). I would recommend that you consider using perceptually uniform colormaps such as those from cmocean (https://matplotlib.org/cmocean/) or Scientific Colour Maps (http://www.fabiocrameri.ch/colourmaps.php). The colormap in Fig. 2a, b is the only one in the paper that seems reasonably perceptually uniform. I believe these colormaps are available in a format that can be imported into ParaVeiw, the tool that I'm pretty sure you are using for this visualization.

I submitted my review well after Reviewer #1's review became available and I feel the need to reiterate a point that she or he made. I fully agree that the paper does not sufficiently discuss the implications of changing heat- and salt-transfer coefficients to vary melt rates. Previous work, cited in this manuscript, have adjusted these coefficients and explored the sensitivity of AS and BS melting to these parameters. But these previous simulations did not, in my understanding, use adjustment of these parameters to change melt rates as a proxy for physical changes in the ocean state (e.g. ocean warming or thermocline shoaling). The implications of using parameter tuning to force melting needs some more discussion. One part of this discussion could presumably be that this approach makes it possible to explore changes to the ocean state (reduced salinity in this case) without complicating the simulation with other changes in state (e.g. changes in surface forcing) that would also impact the ocean state.

A small note: The Cryosphere no longer requires, at least to the best of my knowledge, that the figures and tables be placed at the end of the text during the review process. My request for future manuscripts would be that you include the figures in the text during review and move them to the end only at the point where typesetting occurs (if requested). I review manuscripts electronically and flipping back between the text and the figures and captions gets quite tedious, even more so when I also have to flip back and forth between the main text and the supplement.

**Specific Comments:**

l. 60: "the coefficients are chosen following previous studies": The values for these coefficients are never explicitly stated.

l. 61: "while they are set to 3-times larger values for the CTRL case": As mentioned above, it isn't clear why you chose this to be the control. If this was chosen because melt rates match observations better than for your other simulations, it would be important to state this. Also, as Reviewer #1 points out, it would be somewhat troubling if these larger values of the coefficients are required to compensate for a cooler-than-observed ocean state in this region. If this is the case, it would be worthy of discussion if not, it would be worth discussing why the values used in previous simulations are not the appropriate ones in this case.

l. 62: "is a convenient way to force the ocean model": As I mentioned in the general comments, I think this approach is okay for showing the sensitivity of melting to unknown parameters but shouldn't be treated as an easy substitute for ocean warming, increased inflow of CDW, thermocline shoaling, etc. This needs some more discussion either here or in the discussion section.

l. 71: "(Mazloff et al., 2010; Renault et al., 2011)": Could you quote the observed values (preferably with uncertainties) from these sources? Otherwise, it's hard for the unacquainted TC reader to know how reasonable FESOM's Drake Passage transport is.

l. 73-74, 76: "The bottom temperature on the continental shelf is mostly close to the freezing point except for regions with CDW intrusions onto the AS and BS continental shelves (Figs. 2 and S1)": I guess Fig. S1 is included here because a reader could be expected to deduce from C in Fig. 2 and C - L in Fig. S1 what L would look like, but this seems a little too indirect to me. I would remove the reference to Fig. S1. Similarly for the reference to Fig. S1 on l. 76.

l. 76: "These features are present both in the observations and the model results": The salinity gradient you talk about in AS and BS seems to me to be much more visible in the model results than the observations.

Also, it seems like this is a good place for a discussion of features are not being captured well by the model and what their implications might be.

l. 87: I think Reviewer #1 may have also pointed this out, but Fig. 3 shows bottom salinity, so it's a bit confusing that the text refers to this figure when comparing salinity at 200 m depth.

l. 101: "Despite [being] underestimated by ~50% in magnitude..": Are these really underestimated? You're simulating a different process, changing coefficients in the melt parameterization that lead to an instantaneous jump in meltwater production, than the melt increase seen in observations, so would you expect quantitative agreement with observations? I would reword this to make clear that the process is different and the additional meltwater you see simply is about half of that seen over the last 50 years without stating anything about underestimating.

l. 108-109: "...introduced with heat and salt transfer coefficients being set to 2-times and

30-times larger values, respectively." If you continue to use your current CTRL, these values shoudl be compared with it instead of LMELT (i.e. ⅔ and 10 times the CTRL values, respectively). However, you are clearly treating LMELT as the control experiment throughout this section.

l. 111: "We subtract the LMELT results from MMELT, CTRL, and HMELT": Again, you are using LMELT as the control run.

l. 114-115: "For MMELT-LMELT, the salinity decrease is confined mostly to the AS, BS, and RS continental shelves with a freshening of 0.025 g kg −1 and 0.0030 g kg −1 for the RS continental shelf and RS bottom basin, respectively (Table S4)." This seems like a restatement of the table. As stated below, this table would make more sense if it were moved into the text and the text were modified to provide a broader explanation of the implications of these numbers rather than just restating them.

l. 118, 120: "...WS with values of 0.045, 0.0048, 0.0078, and 0.0035 for the RS shelf..." and "...CTRL case amounting to 0.14, 0.0015, 0.035, and 0.016 for the RS shelf...": These numbers need units and would be better left in the table rather than repeated in the text, as in my previous comment. Instead, the text should presumably discuss the implicaitons of these numbers in a more qualitative way.

l. 132: "could be strongly affected": Could you please elaborate on what you mean by "strongly affected"? What would the effects be? The discussion is currently rather thin but expanding on these effects would help to flesh it out.

l. 139-140 and the remaining paragraph: "We also note that magnitudes of freshening caused by glacial meltwater from ice shelves in the AS and BS **represent linear and nonlinear behaviors**." I think this end phrase is unnecessarily vague. It would be much better, and would flesh out the discussion more, if you discussed what these linear and nonlinear behaviors are rather than simply calling them behaviors. You go on to state the qunatitative amounts of melting under various conditions (again, numbers better left to a table) but you do not explain clearly what the linear behaviors are and differentiate them from the nonlinear ones, nor do you explore why some are linear while others are nonlinear. The discussion section is not an appropriate place for the long lists of numbers you have here. These should only be in tables, and should referred to in the results section, whereas the discussion section should focus on a more qualitative synthesis of the results and their broader implicaitons.

l. 159-160: "We also show that magnitudes of freshening caused by glacial meltwater from the AS and BS represent linear and nonlinear behaviors": This statement isn't very useful to the reader. Can you talk about what these linear and nonlinear behaviors are?

l. 164-165, 174-175: "upon request", "The model code, processing tools, and raw model output are difficult to make publicly available, and the authors recommend contacting the corresponding author for those interested in accessing the data.": I would *strongly* encourage you to work out the logistics of making the specific code (not just a repository but

a specific DOI on a site like Zenodo) available. Having code available only on request really hampers open science and model development. Not recording in the paper exactly which version of the code was used further hampers reproducability. I realize this is a bit of extra work and sometimes requires getting premission from the developers but it is worth the effort to the broader community. I ask you to reconsider.

I realize that data sets are harder to make available but I would encourage you to see if a database like Pangaea, Open Science Framework or the Earth System Grid might be an appropriate place to host your data in a public forum. Again, the lack of data availability really sets back open science.

Figs. 1, 2c,d, 3, 4, and 5: As mentioned in the general comments, I would encourage the authors to look at alternative, perceptually unifrom colormaps.

Fig. 2a, b: If you end up switching to using LMELT as the reference simulation as I have recommended, it may be more appropriate to plot that one here. Also, the obs. plots are really tiny and hard to compare. It would be helpful to have bias plots (CTRL - Obs) in addition.

Fig. 2c, d: It would likely make sense to move these panels to another figure to make more room for expanding the first two panels of this figure.

Fig. 3: As in the text, it seems like you are treating the LMELT as the control run here, since differences are C-L, not L-C.

Table S1: It seems like the ice shelf-ocean drag coefficient is missing. I would also suggest putting the control values of the heat- and salt-transfer coefficients here. Are there any other parameter values related to the ice shelf-ocean interface or boundary layer that you did not include? Is there a good reason this should be in the supplement rather than the main text? I don't have strong feelings either way, but as a reader I often don't download the supplement if I don't need to.

Table S2: This data seems too simple to be worth having a table, and the text repeated in each entry seems unnecessary.

Table S3: As Reviewer 1 pointed out, references 16 and 17 need to be replaced with the proper reference or shortcuts of some kind. Maybe these were the numbers of the references when this paper was submitted to another journal with another citation format?

Table S4: This table belongs in the main manuscript. This is also a case where LMELT, rather than CTRL, seems to be treated as the control run. The formatting of the leftmost column is hard to follow as the spacing between adjacent lines in the same entry and between table rows is the same (hopefully typesetting will fix this).

Figs. S1 and S2: Once again, by taking CTRL - LMELT rather than the other way around, it seems like LMELT is the control run.

**Typographical and grammatical corrections:**

l. 13: "based on satellite-based" sounds a bit redundant so I'd suggest changing "based on" to something like "as shown by"

l. 19: "Their" probably refers to ice shelves in AS and BS, but this is not entirely clear from the context, so I would make this explicit.

l. 19: The comma after "observations" should be removed.

l. 21: "evidences" should be something like "lines of evidence"

l. 31: "Kusahara and Hasumi (2014); Dinniman et al. (2016); Kusahara et al. (2017)": I don't think explicit ("citet" in LaTex) citations should be combined in this way. I would reword the sentence so these become parenthetical citations, e.g. "Using circum-Antarctic or global domains, several studies (Kusahara and Hasumi 2014; Dinniman et al. 2016; Kusahara et al. 2017) also showed…" If you want to keep them more as they are, I think you need to put "and" before "Kusahara et al.".

l. 34: "are developed" should be "have been devleoped"

l. 38: "(FESOM) (Timmermann..." should be "(FESOM; Timmermann…"

l. 61: "3-times" should be "three times"

l. 76: "both in" should be "in both"

l. 80-81: "(Depoorter et al., 2013; Rignot et al., 2013) (Table S3)" might be more cleanly formatted as "(Table S3; Depoorter et al., 2013; Rignot et al., 2013)" or by rephrasing so that the satellite estimates are mentioned earlier (with references) than the CTRL results.

l. 83: "LMELT case" should either be "the LMELT case" or just "LMELT". I would suggest rephrasing this whole sentence: "...may represent better the melt rates  in the middle of the last century"

l. 84: "largely" should be something like "significantly"

l. 85: "flown" should be "flowed"

l. 89-90: The citations would be better formatted as "(Fig. 2; Nakayama et al., 2014; Dinniman et al., 2016)"

l. 97: "RS dense shelf water observed for about 50 years shows" should be something like "Fifty years of observations of RS dense shelf water show..."

l. 99: "(RSBW) (Purkey and Johnson, 2013)" should be "(RSBW; Purkey and Johnson, 2013)", although this is confusing since the citation isn't about RSBW but rather its warming and freshending so it might be best to reword the sentence so the citations and the abbreviation can be separated.

l. 100: "RSBW shows warming and freshening of ~0.1 ◦ C and ~0.01 g kg −1 , respectively" mighty be better as "RSBW experiences a ~0.1 ◦ C warming and a ~0.01 g kg −1 freshening".  (Sorry for the formatting.)

l. 101: "Despite underestimated by ~50% in magnitude": something is missing here and this should maybe be "Despite being underestimated by ~50% in magnitude"

l. 107: "focusing on both small (200-m depth) and large (bottom) depths": I find this wording confusing and I think it would work just as well as "focusing on both 200 meters depth and the sea floor" (or "ocean bottom" if you prefer).

l. 108: "introduced with heat and salt transfer coefficients  set to..."

l. 132: no comma is needed after "affected"

l. 134-135: "...the idea presented by (e.g. Beckmann and Timmermann, 2001)": This should be  "...the idea presented by e.g. Beckmann and Timmermann (2001)".  You might want "...by, e.g., Beckmann…" but I don't think the commas are required.

l. 137-138: "However, considering the magnitude of the salinity decrease in the CTRL experiment, circum-Antarctic freshening could possibly be  be underway."

l. 141: "enhances" should be "is enhanced"

l. 154: "existing" would be better as somthing like "recent"

l. 155: "We further show...propagates further downstream": It's a little jarring to have "further" twice in this sentence so I'd suggest starting with something like "In addition, we show"

l. 166: "All authors commented on the manuscript."

---

## Referee Comment (RC4) · Xylar Asay-Davis (Referee) · 18 Jan 2020

Xylar Asay-Davis (Referee)

xylar@lanl.gov

I realized just after submitting my review that I had not completed the first 3 bullet points, which should have read:

This manuscript explores the impacts of changes in freshwater fluxes from the Amundsen Sea (AS) and Bellingshausen Sea (BS) on the Ross continental shelf, depper Ross Sea (RS) and other Antarctic regions. Major findings are that:

* Freshwater reaches the Ross continental shelf in one year, the deeper Ross Sea within ∼5 years, the region near the Amery Ice Shelf after ∼5-10 years and the Weddell Sea in ∼10-15 years.

* For the most significant amounts of melting, on the order of 10 times currently ob-

served melt rates in the AB region, freshwater reaches the Weddell Sea much more quickly (∼10 years into the simulation) and the amount of freshwater reaching the Weddell continental shelf is enough to reduce the salinity there by a non-negligible amount.

\* In simulations with AS and BS melt rates comparable to or less than present-day, meltwater may reach the Weddell Sea after ∼30 years but its impact on salinity are difficult to distinguish from temporal (and perhaps ensemble) variability

Again, my apologies
* * *

---

## Author Comment (AC1) · 31 Mar 2020

*Response to the specific comments and corrections from the Editors*
*(Comments from reviewer are in italics; our responses are indicated in bold typeface)*

*Reviewers' comments:*
*Reviewer #1 (Remarks to the Author):*
*In this study the authors use the finite element ocean/sea ice/ice shelf FESOM model to study the impact of increased basal melting of the ice shelves in the Amundsen (AS) and Bellingshausen (BS) Seas on the hydrography of the entire Antarctic continental shelf and the condition of Antarctic Bottom Water (AABW) just off the continental shelf. This is done through examining four 32 year long simulations where the ice shelf basal melt rates are increased between simulations by modifying the transfer coefficients between the ice and the water underneath. The freshening signal not only propagates onto the Ross Sea continental shelf, but within the time frame of these simulations also makes its way around almost the entire continent onto the Weddell Sea continental shelf. The increased melt scenarios also impact the AABW off the Ross Sea and have slight impacts on AABW elsewhere. I thought the paper was generally clear and well written. The impacts of increasing ice shelf basal melt in the AS and BS on Antarctic continental shelf waters and AABW are an important problem and, in my opinion, well worth the attention of the Cryosphere. FESOM, with its high resolution on the Antarctic continental shelf (including under the ice shelf cavities) and slope, explicit ice shelves, and global domain (so no worries about lateral boundary conditions) is a fantastic tool to study this question.*

*My only negative general comment is relatively minor, but I do think there are some warnings about the applicability of these results that should be included. The LMELT results are using what the authors think the heat and salt transfer coefficients should be, but this results in low basal melting compared to observations. No mention is made of why they think the melting is low compared to current day conditions: Is this because of unknown ice/ocean interaction physics or is there a problem with the representation of water masses on the continental shelf? If it's an issue with the water masses, does this influence the rate at which meltwater advects (due to lateral density gradients) in the coastal current over the continental shelf? Also, the HMELT increased melting is not due to changes in the AS/BS shelf conditions, as they presumably are in the real world since the mid-20th century, but rather numerical manipulation of the ice/ocean transfer coefficients. Does this have an impact on the results?*

**In the revised manuscript, we include a discussion explaining the reason why the melting is low compared to observations for LMELT. We also elaborate on the physical meaning of changing turbulent heat and salt transfer coefficient (Lines 146-167).**

*I have some other specific comments and suggestions below, but most of these are very minor and should be easily dealt with by the authors.*

*Specific comments*
*Abstract, line 3: The abstract states that the long term impact of enhanced melting of the Amundsen Sea ice shelves "on the Southern Ocean hydrography has not been well investigated". However, there have been several studies of this (e.g. Fogwill et al., 2015; Golledge et al., 2019; Lago and England, 2019), just not with models setup as nicely as the FESOM model used here (i.e. explicit ice shelves and high resolution around Antarctica). I think it would be helpful to mention some of the other studies in the Introduction, but also include mention of why the model used here is better suited for examining this question.*
**Revised as suggested (Line 41-43).**

*Abstract, line 7 and line 155: See comments below about the propagation of the melt- water, but suggest changing "propagates further" to "can propagate further".*
**Revised as suggested.**

*First paragraph of model section: Even though the authors mention the ice/ocean heat/salt transfer coefficients in the next paragraph, I think it would be helpful to readers not familiar with FESOM to explicitly mention in this paragraph that FESOM does simulate the melting/freezing of the base of the floating ice shelves.*
**Revised as suggested.**

*Should also explicitly mention that FESOM does have a dynamic sea ice model.*
**Revised as suggested.**

*Lines 71-73: I think it would be helpful if the authors added a figure about the simulated sea ice extent to the Supplement.*
**As suggested, we included a figure of simulated sea ice extent (Fig. S1).**

*Lines 124-126: The HMELT case shows the propagation of the freshening signal as described here, but it's often hard to see if there has been a propagation of the signal in the other cases. For example, the red and orange lines in Figure 6c do not stay below zero until ~ year 15 and then go back above zero for a good portion of the time past year 20. In 6d, one could argue that the red and orange lines do not stay below zero until almost the end of the period. This is why I suggested the change in line 7 of the Abstract/line 155.*

**Thank you for your suggestion and we revised the manuscript as suggested.**

*Line 151: I think it's a bit much to say this paper is investigating the impact of the meltwater on "the Southern Ocean hydrography". It is looking at some aspects of the hydrography (Antarctic continental shelf conditions and changes in AABW), but not at all the broad scale water masses that are involved in the Southern Ocean. Suggest changing "Southern Ocean hydrography" to something a little more focused.*
**Revised as suggested (Line 189).**

*Technical corrections*
*Line 15: To avoid confusion from some readers about ice shelf vs. grounded ice contributions to sea level rise, suggest changing "ongoing sea level rise and ocean freshening" to "ongoing ocean freshening as well as to sea level rise".*
**Revised as suggested.**

*Line 21: Suggest changing "There exist a few other evidences" to "There is some evidence".*
**As suggested by reviewer we replace "a few other evidences" with "a few lines of evidence"**

*Line 35: Suggest changing "focuses" to "of a focus".*
**Revised as suggested.**

*Lines 60 and 61: Are the transfer coefficients set to constants as in Hellmer and Olbers or functions of the friction velocity as in Holland and Jenkins? From other FESOM ice shelf papers, I assume they are functions of the friction velocity, but I can't tell from how it is written here.*
**We calculated the ice shelf melt rate following Holland and Jenkins 1999. We revised the manuscript as in Line 67-68.**

*Line 82: From Rignot et al. (2013), I get 664 Gt/yr (not 459) for their estimate of the basal melt of the combined AS and BS (numbers 5-18 in Table S3) ice shelves.*
**This is from Supplementary Table in Rignot et al., 2013. This is the steady state melt rate ($B_{SS}$) assuming zero thickening or thinning. My calculation was based on the table I received from personal communication and recalculated the number (461 Gt/yr) based on their publication. To clarify, we revised as the manuscript (Line 96-97).**

*Line 83: Change "at that the time in the middle" to "in the middle".*
**Revised as suggested.**

*Lines 87-88 and Figure 3: If the Figure 3 plots are mean bottom salinity, then how does this show that the salinity at 200-m depth is stable? Is "bottom" over the continental shelf in the figure defined at 200-m?*

**Thank you for pointing this out. We removed this sentence in the revised manuscript.**

*Line 89: Suggest changing "the RS continental shelf further along the east Antarctic coast and towards" to "the RS continental shelf and then further along the east Antarctic coast as well as towards".*

**Revised as suggested.**

*Line 92: Typo, "Fig .3" should be "Fig. 3".*

**Revised as suggested.**

*Line 101: Suggest changing "Despite underestimated" to "Despite being underestimated".*

**This sentence is removed in the revised manuscript.**

*Line 143: Typo, "0.030" should be "0.0030" and "0.048" should be "0.0048" (assuming Table S4 is correct).*

**Revised as suggested.**

*Line 167: Add "on" after "commented".*

**Revised as suggested.**

*Figure 2: Why does the temperature scale top out at 1.0C? The Schmidtko et al. observations have the mean BS temperature > 1.0, and thus it's hard to make comparisons between the model and the observations in the AS and BS continental shelves.*

**In the revised manuscript, we decided to remove this figure. Instead, we included a few sentences describing oceanographic features well captured and not well captured in the model simulation (Line 84-90)**

*Table S1: What are the units for the sea ice salt concentration and is the value here correct? Timmermann et al. (2009) has it as 5 (psu or g/kg).*

**Thank you for pointing this out. This value here is not correct and sea ice salt concentration is 5 (g/kg) as pointed out.**

*Table S3: I don't understand what "16" and "17" are in the references. I assume one is Depoorter et al. and one is Rignot et al., but can't tell which is which.*

**Thank you for pointing this out. We revised the manuscript as suggested.**

*Reviewer #2 (Remarks to the Author):*
*Many satellite and oceanographic observations revealed that Antarctic Ice sheets and the Southern Ocean have been changing in recent decades. The interaction between Antarctic ice sheets/shelves and the Southern Ocean is one of the most important topics in the climate sciences. This study investigates pathways of ice-shelf meltwater from the West Antarctic ice shelves and its role on the Southern Ocean conditions, using a series of sea-ice/ice-shelf/ocean simulations. In my reading, the results of this study will be a valuable contribution to the Antarctic sciences. I recommend publication in The Cryosphere after addressing the comments listed below.*

**Thank you very much for encouraging and insightful comments.**

*I have two major comments.*

*(1) This paper used numerical experiments with different levels of basal melting (by tuning the transfer coefficients) to explain the observed changes (e.g., lines 97-104). In my reading, the results from CTRL (or CTRL-LMELT) shows a transition from the LMELT conditions and are not suitable for explaining the observed changes. The transition timescale is useful information, but the comparison of the Southern Ocean water properties between the model and observation in the present manuscript may be misleading.*

**We think that PRS-LMELT (previously CTRL-LMELT) possibly explains some of the observed changes. We revised the manuscript as in (Line 147-167). We also decided to remove figures comparing data and observations. We now show the LMELT case rather than the PRS case and we think it is sufficient to cite Schmitko et al., 2014 for comparison.**

*(2) Although there are sentences about the impact of the meltwater on AABW formation in the remote regions (Cape Darnley and Weddell Sea) in abstract and discussion (lines 9-11 and158-159), Figure 5d-f show no pronounced change in the bottom water properties. I understand the idea, but the simulations didn't support it.*

**I agree that our results do not fully support this idea. We revised the manuscript as in Line 11-12 and 197-198.**

*Specific comments:*

*(3) lines 31-34: Wrong and missing citations Kusahara et al. (2017) is a modeling study of dense shelf water, not ice-shelf meltwater. Kusahara and Hasumi (2013, JGR-Oceans) performed virtual (meltwater) tracer experiments in idealized warming climates, showing that increased basal meltwater from the Amundsen and Bellingshausen Seas causes the bottom water freshening in the Ross Sea and Australia- Antarctic Basins.*

**Thank you for pointing out. We removed Kusahara et al., 2017. We also included Kusahara and Hasumi 2013 and state that virtual (meltwater) tracer experiments in idealized warming climates show that increased basal meltwater from the Amundsen and Bellingshausen Seas causes the bottom water freshening in the Ross Sea.**

*(4) lines 38-40 Please briefly explain what kind of model development allows the longer integration.*
**Revised as suggested (Line 44-45).**

*(5) The description of ice-ocean interaction is missing.*
**We revised the manuscript as in Line 62-64.**

*(6) I think that 10-years spin-up is short.*
**We revised the manuscript as in Line 71-72.**

*(7) Lines 97-104 and Fig. S2a What is the mechanism of the bottom water warming in the Ross Sea?*
**We revised the manuscript as in Line 120.**

*(8) All map figures need longitude and latitude information (at least one panel).*
**We included lat-lon information on Figure 1.**

*(9) The manuscript is not so long. I suggest merging the supplementary material into the main text to increase readability.*
**As suggested, we moved all the tables to the revised manuscript but kept a few figures in the supplementary.**

*Reviewer #3 (Xylar Davis )*
*I realized just after submitting my review that I had not completed the first 3 bullet points, which should have read:*

*This manuscript explores the impacts of changes in freshwater fluxes from the Amund- sen Sea (AS) and Bellingshausen Sea (BS) on the Ross continental shelf, depper Ross Sea (RS) and other Antarctic regions. Major findings are that:*

*\* Freshwater reaches the Ross continental shelf in one year, the deeper Ross Sea within ~5 years, the region near the Amery Ice Shelf after ~5-10 years and the Weddell Sea in ~10-15 years.*

*\* For the most significant amounts of melting, on the order of 10 times currently observed melt rates in the AB region, freshwater reaches the Weddell Sea much more quickly (~10 years into the simulation) and the amount of freshwater reaching the Weddell continental shelf is enough to reduce the salinity there by a non-negligible amount.*

*\* In simulations with AS and BS melt rates comparable to or less than present-day, meltwater may reach the Weddell Sea after ~30 years but its impact on salinity are difficult to distinguish from temporal (and perhaps ensemble) variability*

*General Comments:*

*Main points:*

*- Freshwater from the Amundsen and Bellingshausen (AB) Seas is shown to reach the Ross continental shelf in XXX years, the deeper Ross Sea within XXX years, the region near the Amery Ice Shelf after XXX years and the Weddell Sea in XXX years.*

*- For the most significant amounts of melting, on the order of 10 times currently observed melt rates in the AB region, freshwater reaches the Weddell Sea much more quickly (~10 years into the simulation) and the amount of freshwater reaching the Weddell continental shelf is enough to reduce the salinity there by a non-negligible amount.*

*- In simulations with AB melt rates comparable to or less than present-day, meltwater may reach the Weddell Sea after ~30 years but its impact on salinity are difficult to distinguish from temporal (and perhaps ensemble) variability*

*- I get the impression in several places in the text that the experiments were designed (and perhaps the manuscript was originally written) with LMELT as the intended control experiment, and that perhaps a later decision was made that CTRL should be the control because its melt rates are most comparable to observations. Differences are repeatedly taken with respect to LMELT, rather than CTRL and the coefficients used in LMELT are stated to be the ones commonly used in other FESOM simulations. I would urge you to consider explicitly renaming LMELT to CTRL and CTRL to something else because this would seem more consistent with the manuscript as written. Several specific comments point out in more detail where this inconsistency arises.*

**Following the reviewer's suggestion, we decided to consider LMELT as a reference simulation. The CTRL case is now renamed as PRS because the latter represents ice shelf**

**melt rates closer to the current observations. We also state that we regard LMELT as the reference simulation. The manuscript is revised as Line 69-70.**

*The manuscript seems short for TC, especially the discussion section (see below). It sometimes reads as if it were intended for a journal that requires a shorter page count. This may explain why several tables that are referenced repeatedly in the text are included in the supplement rather than the main manuscript. I am not aware of a limit on tables or figures that require putting so many tables in the supplement. In particular, values from Table S4 are repeated (in multiple places) in the text, which would presumably not be necessary if that table were in the main text.*
**Following the reviewer's suggestion, we moved all tables to the main text.**

*Speaking of which, there is a lot of redundancy both within the text vs. in tables and between the results, discussion, and conclusion sections. I have pointed out where I find this redundancy in the specific comments. This redundancy comes at the expense of what could have been a broader discussion of the results that synthesizes the findings in a somewhat more qualitative fashion and talks about their broader implications based on observed and projected changes in AB melting, impacts of freshening on both the Ross continental shelf, deeper Ross Sea and elsewhere, etc.*
**Following the reviewer's suggestion in the specific comments, we moved all tables to the main text. See other responses to the specific comments.**

*Colormaps are not very intuitive and are not friendly to readers with color blindness. The manuscript preparation guidelines include the following: "For maps and charts, please keep colour blindness in mind and avoid the parallel usage of green and red. For a list of colour scales that are illegible to a significant number of readers, please visit ColorBrewer 2.0." In addition to concerns about color blindness, the colormaps used in this manuscript suffer from alternate banding of bright and dark colors that make it difficult for a reader to intuitively tell higher from lower values of the field. (In the terminology of color theory, they are not perceptually uniform). I would recommend that you consider using perceptually uniform colormaps such as those from cmocean (https://matplotlib.org/cmocean/) or Scientific Colour Maps (http://www.fabiocrameri.ch/colourmaps.php). The colormap in Fig. 2a, b is the only one in the paper that seems reasonably perceptually uniform. I believe these colormaps are available in a format that can be imported into ParaVeiw, the tool that I'm pretty sure you are using for this visualization.*
**In the revised manuscript, we only use perceptually uniform colormaps.**

*I submitted my review well after Reviewer #1's review became available and I feel the need to reiterate a point that she or he made. I fully agree that the paper does not sufficiently discuss the*

*implications of changing heat- and salt-transfer coefficients to vary melt rates. Previous work, cited in this manuscript, have adjusted these coefficients and explored the sensitivity of AS and BS melting to these parameters. But these previous simulations did not, in my understanding, use adjustment of these parameters to change melt rates as a proxy for physical changes in the ocean state (e.g. ocean warming or thermocline shoaling). The implications of using parameter tuning to force melting needs some more discussion. One part of this discussion could presumably be that this approach makes it possible to explore changes to the ocean state (reduced salinity in this case) without complicating the simulation with other changes in state (e.g. changes in surface forcing) that would also impact the ocean state.*

**In the revised manuscript, we included a more comprehensive discussion (Line 146-167).**

*A small note: The Cryosphere no longer requires, at least to the best of my knowledge, that the figures and tables be placed at the end of the text during the review process. My request for future manuscripts would be that you include the figures in the text during review and move them to the end only at the point where typesetting occurs (if requested). I review manuscripts electronically and flipping back between the text and the figures and captions gets quite tedious, even more so when I also have to flip back and forth between the main text and the supplement.*

**I consider changing the format for future submissions. For this submission, however, I will keep the same format and, thus, page numbers as it can be easily compared with the previous submission. Thank you for your suggestion.**

*Specific Comments:*
*I. 60: "the coefficients are chosen following previous studies": The values for these coefficients are never explicitly stated.*

**We calculated ice shelf melt rate based on Holland and Jenkins 1999. The only difference is that the drag coefficient at the ice shelf base is set to $2.5*10^{-3}$. We emphasize this point in the revised manuscript.**

*I. 61: "while they are set to 3-times larger values for the CTRL case": As mentioned above, it isn't clear why you chose this to be the control. If this was chosen because melt rates match observations better than for your other simulations, it would be important to state this.*

**As suggested, we now use LMELT as our reference simulation.**

*Also, as Reviewer #1 points out, it would be somewhat troubling if these larger values of the coefficients are required to compensate for a cooler-than-observed ocean state in this region. If this*

*is the case, it would be worthy of discussion if not, it would be worth discussing why the values used in previous simulations are not the appropriate ones in this case.*

**We also include additional discussion in the revised manuscript on the choice of turbulent heat and salt transfer coefficients (Line 146-167).**

*l. 62: "is a convenient way to force the ocean model": As I mentioned in the general comments, I think this approach is okay for showing the sensitivity of melting to unknown parameters but shouldn't be treated as an easy substitute for ocean warming, increased inflow of CDW, thermocline shoaling, etc. This needs some more discussion either here or in the discussion section.*

**Please note, the purpose of this paper is to show the impact of increased AS and BS ice shelf basal melting on the hydrography of downstream Antarctic marginal seas and less on the impact of ocean warming, increased on-shore flow and/or thermocline shoaling on basal melt rates in AS and BS. However, we included additional discussion on this point (Line 146-167).**

*l. 71: "(Mazloff et al., 2010; Renault et al., 2011)": Could you quote the observed values (preferably with uncertainties) from these sources? Otherwise, it's hard for the unacquainted TC reader to know how reasonable FESOM's Drake Passage transport is.*

**Revised as suggested.**

*l. 73-74, 76: "The bottom temperature on the continental shelf is mostly close to the freezing point except for regions with CDW intrusions onto the AS and BS continental shelves (Figs. 2 and S1)": I guess Fig. S1 is included here because a reader could be expected to deduce from C in Fig. 2 and C - L in Fig. S1 what L would look like, but this seems a little too indirect to me. I would remove the reference to Fig. S1. Similarly, for the reference to Fig. S1 on l. 76.*

**Revised as suggested.**

*l. 76: "These features are present both in the observations and the model results": The salinity gradient you talk about in AS and BS seems to me to be much more visible in the model results than the observations.*

**As suggested, we note this point in the revised manuscript (Line 87-90).**

*Also, it seems like this is a good place for a discussion of features are not being captured well by the model and what their implications might be.*

**As suggested, we also included a sentence describing what is not being well captured in the model simulation (Line 87-90).**

*l. 87: I think Reviewer #1 may have also pointed this out, but Fig. 3 shows bottom salinity, so it's a bit confusing that the text refers to this figure when comparing salinity at 200 m depth.*

**This sentence is now removed.**

*l. 101: "Despite [being] underestimated by ~50% in magnitude..": Are these really underestimated? You're simulating a different process, changing coefficients in the melt parameterization that lead to an instantaneous jump in meltwater production, than the melt increase seen in observations, so would you expect quantitative agreement with observations? I would reword this to make clear that the process is different and the additional meltwater you see simply is about half of that seen over the last 50 years without stating anything about underestimating.*

**We revised the manuscript as suggested (Line 112-120)**

*l. 108-109: "...introduced with heat and salt transfer coefficients being set to 2-times and 30-times larger values, respectively." If you continue to use your current CTRL, these values should be compared with it instead of LMELT (i.e. 2/3 and 10 times the CTRL values, respectively). However, you are clearly treating LMELT as the control experiment throughout this section.*

**Following the reviewer's suggestion, we now use LMELT as our reference simulation.**

*l. 111: "We subtract the LMELT results from MMELT, CTRL, and HMELT": Again, you are using LMELT as the control run.*

**See previous reply.**

*l. 114-115: "For MMELT-LMELT, the salinity decrease is confined mostly to the AS, BS, and RS continental shelves with a freshening of 0.025 g kg −1 and 0.0030 g kg −1 for the RS continental shelf and RS bottom basin, respectively (Table S4)." This seems like a restatement of the table. As stated below, this table would make more sense if it were moved into the text and the text were modified to provide a broader explanation of the implications of these numbers rather than just restating them.*

**As suggested, we moved this table into the main text. Although we still keep these numbers in the results section, we modified the text to provide a broader explanation (Line 168-186).**

*l. 118, 120: "...WS with values of 0.045, 0.0048, 0.0078, and 0.0035 for the RS shelf..." and "...CTRL case amounting to 0.14, 0.0015, 0.035, and 0.016 for the RS shelf...": These numbers need units and would be better left in the table rather than repeated in the text, as in my previous comment. Instead, the text should presumably discuss the implications of these numbers in a more qualitative way.*

**Revised as suggested.**

*l. 132: "could be strongly affected": Could you please elaborate on what you mean by "strongly affected"? What would the effects be? The discussion is currently rather thin but expanding on these effects would help to flesh it out.*
**Revised as suggested (Line 181-186).**

*l. 139-140 and the remaining paragraph: "We also note that magnitudes of freshening caused by glacial meltwater from ice shelves in the AS and BS represent linear and nonlinear behaviors." I think this end phrase is unnecessarily vague. It would be much better, and would flesh out the discussion more, if you discussed what these linear and nonlinear behaviors are rather than simply calling them behaviors. You go on to state the quantitative amounts of melting under various conditions (again, numbers better left to a table) but you do not explain clearly what the linear behaviors are and differentiate them from the nonlinear ones, nor do you explore why some are linear while others are nonlinear. The discussion section is not an appropriate place for the long lists of numbers you have here. These should only be in tables, and should referred to in the results section, whereas the discussion section should focus on a more qualitative synthesis of the results and their broader implications.*
**Revised as suggested (Line 169-180).**

*l. 159-160: "We also show that magnitudes of freshening caused by glacial meltwater from the AS and BS represent linear and nonlinear behaviors": This statement isn't very useful to the reader. Can you talk about what these linear and nonlinear behaviors are?*
**Revised as suggested (Line 197-200).**

*l. 164-165, 174-175: "upon request", "The model code, processing tools, and raw model output are difficult to make publicly available, and the authors recommend contacting the corresponding author for those interested in accessing the data.": I would strongly encourage you to work out the logistics of making the specific code (not just a repository but a specific DOI on a site like Zenodo) available. Having code available only on request really hampers open science and model development. Not recording in the paper exactly which version of the code was used further hampers reproducability. I realize this is a bit of extra work and sometimes requires getting premission from the developers but it is worth the effort to the broader community. I ask you to reconsider. I realize that data sets are harder to make available but I would encourage you to see if a database like Pangaea, Open Science Framework or the Earth System Grid might be an appropriate place to host your data in a public forum. Again, the lack of data availability really sets back open science.*

**As suggested, we upload grid data and model output into a server in our institute.**

*Figs. 1, 2c,d, 3, 4, and 5: As mentioned in the general comments, I would encourage the authors to look at alternative, perceptually unifrom colormaps.*
**Revised as suggested.**

*Fig. 2a, b: If you end up switching to using LMELT as the reference simulation as I have recommended, it may be more appropriate to plot that one here. Also, the obs. plots are really tiny and hard to compare. It would be helpful to have bias plots (CTRL - Obs) in addition.*
**As we would like to show that the model is capable of simulating the hydrographic spatial structures in consistent with observations, we decided to remove observations from Figure 2. Instead, we cite these figures in the manuscript and further include a sentence discussing what features the model can and cannot reproduce (Line 87-90).**

*Fig. 2c, d: It would likely make sense to move these panels to another figure to make more room for expanding the first two panels of this figure.*
**Revised as suggested.**

*Fig. 3: As in the text, it seems like you are treating the LMELT as the control run here, since differences are C-L, not L-C.*
**Following the reviewer's suggestion, we now use LMELT as the reference simulation.**

*Table S1: It seems like the ice shelf-ocean drag coefficient is missing. I would also suggest putting the control values of the heat- and salt-transfer coefficients here. Are there any other parameter values related to the ice shelf-ocean interface or boundary layer that you did not include? Is there a good reason this should be in the supplement rather than the main text? I don't have strong feelings either way, but as a reader I often don't download the supplement if I don't need to.*
**As suggested, we now include the value of ice shelf-ocean drag coefficient in this table. Other parameters are the same as Holland and Jenkins 1999. We also move this table to the main text.**

*Table S2: This data seems too simple to be worth having a table, and the text repeated in each entry seems unnecessary.*
**This table is removed.**

*Table S3: As Reviewer 1 pointed out, references 16 and 17 need to be replaced with the proper reference or shortcuts of some kind. Maybe these were the numbers of the references when this paper was submitted to another journal with another citation format?*

**Revised as suggested.**

*Table S4: This table belongs in the main manuscript. This is also a case where LMELT, rather than CTRL, seems to be treated as the control run. The formatting of the leftmost column is hard to follow as the spacing between adjacent lines in the same entry and between table rows is the same (hopefully typesetting will fix this).*

**Revised as suggested.**

*Figs. S1 and S2: Once again, by taking CTRL - LMELT rather than the other way around, it seems like LMELT is the control run.*

**We now use LMELT as the reference simulation.**

*Typographical and grammatical corrections:*
*l. 13: "based on satellite-based" sounds a bit redundant so I'd suggest changing "based on" to something like "as shown by"*

**Revised as suggested.**

*l. 19: "Their" probably refers to ice shelves in AS and BS, but this is not entirely clear from the context, so I would make this explicit.*

**Revised as suggested.**

*l. 19: The comma after "observations" should be removed.*

**Revised as suggested.**

*l. 21: "evidences" should be something like "lines of evidence"*

**Revised as suggested.**

*l. 31: "Kusahara and Hasumi (2014); Dinniman et al. (2016); Kusahara et al. (2017)": I don't think explicit ("citet" in LaTex) citations should be combined in this way. I would reword the sentence so these become parenthetical citations, e.g. "Using circum-Antarctic or global domains, several studies (Kusahara and Hasumi 2014; Dinniman et al. 2016; Kusahara et al. 2017) also showed..." If you want to keep them more as they are, I think you need to put "and" before "Kusahara et al.".*

**Revised as suggested.**

*l. 34: "are developed" should be "have been devleoped"*
**Revised as suggested.**

*l. 38: "(FESOM) (Timmermann..." should be "(FESOM; Timmermann..."*
**Revised as suggested.**

*l. 61: "3-times" should be "three times"*
**Revised as suggested.**

*l. 76: "both in" should be "in both"*
**Revised as suggested.**

*l. 80-81: "(Depoorter et al., 2013; Rignot et al., 2013) (Table S3)" might be more cleanly formatted as "(Table S3; Depoorter et al., 2013; Rignot et al., 2013)" or by rephrasing so that the satellite estimates are mentioned earlier (with references) than the CTRL results.*
**Revised as suggested.**

*l. 83: "LMELT case" should either be "the LMELT case" or just "LMELT". I would suggest rephrasing this whole sentence: "...may represent better the melt rates at that the time in the middle of the last century"*
**Revised as suggested.**

*l. 84: "largely" should be something like "significantly" l. 85: "flown" should be "flowed"*
**Revised as suggested.**

*l. 89-90: The citations would be better formatted as "(Fig. 2; Nakayama et al., 2014; Dinniman et al., 2016)"*
**Revised as suggested.**

*l. 97: "RS dense shelf water observed for about 50 years shows" should be something like "Fifty years of observations of RS dense shelf water show..."*
**Revised as suggested.**

*l. 99: "(RSBW) (Purkey and Johnson, 2013)" should be "(RSBW; Purkey and Johnson, 2013)", although this is confusing since the citation isn't about RSBW but rather its warming and freshending so it might be best to reword the sentence so the citations and the abbreviation can be separated.*

**We removed all the abbreviations for RSBW in the revised manuscript.**

*l. 100: "RSBW shows warming and freshening of ~0.1◦C and ~0.01 g kg −1 , respectively" mighty be better as "RSBW experiences a ~0.1◦C warming and a ~0.01 g kg −1 freshening". (Sorry for the formatting.)*

**Revised as suggested.**

*l. 101: "Despite underestimated by ~50% in magnitude": something is missing here and this should maybe be "Despite being underestimated by ~50% in magnitude"*

**This sentience is removed.**

*l. 107: "focusing on both small (200-m depth) and large (bottom) depths": I find this wording confusing and I think it would work just as well as "focusing on both 200 meters depth and the sea floor" (or "ocean bottom" if you prefer).*

**Revised as suggested.**

*l. 108: "introduced with heat and salt transfer coefficients being set to..."*

**Revised as suggested.**

*l. 132: no comma is needed after "affected"*

**This sentence is removed.**

*l. 134-135: "...the idea presented by (e.g. Beckmann and Timmermann, 2001)": This should be "...the idea presented by e.g. Beckmann and Timmermann (2001)". You might want "...by, e.g., Beckmann..." but I don't think the commas are required.*

**Revised as suggested.**

*l. 137-138: "However, considering the magnitude of the salinity decrease in the CTRL experiment, circum-Antarctic freshening could possibly be possibly undergoing be underway."*

**Revised as suggested.**

*l. 141: "enhances" should be "is enhanced"*

**Revised as suggested.**

*l. 154: "existing" would be better as something like "recent"*

**Revised as suggested.**

*l. 155: "We further show...propagates further downstream": It's a little jarring to have "further" twice in this sentence so I'd suggest starting with something like "In addition, we show"*

**Revised as suggested.**

*l. 166: "All authors commented on the manuscript."*

**Revised as suggested.**

---

## Author Comment (AC4) · 31 Mar 2020

My replies are in pdf attached as supplement.

———————————————————

---

## Author Response (AR2)

Response to the specific comments and corrections from the Editors

(Comments from reviewer are in italics; our responses are indicated in bold typeface)

*Dear authors,*

*thank you for your revisions. Your manuscript has greatly improved. Before proceeding, pls address the following issues in your latest ms version. -- Thank you.*

**Thank you for your comments.**

*l64: Change "We carry out" to "We carried out"*

**Done.**

*l66: Change "(Holland and Jenkins, 1999) but the drag coefficient" to "Holland and Jenkins (1999) with the drag coefficient".*

**Done.**

*l80: Change "Using ocean state" to "Using the ocean state".*

**Done.**

*l83: Change "The time series of sea ice extent also show similar variability to observations" to "The variability of the modelled sea ice extent is similar to those in the observed data".*

**Done.**

*l85: Change "Bottom salinity shows" to "The bottom salinity exhibits".*

**Done.**

*l85: Change "local salinity maxima" to "local maxima".*

**Done.**

*l86: Change "in both" to "in both,".*

**Done.**

*l87: Change "Despite general" to "Despite the general".*

**Done.**

*l88: Change "on-shelf CDW temperature is" to "the on-shelf CDW temperatures are".*

**Done.**

*l88: Change "roughly by 0.5 C" to "by roughly 0.5 C".*

**Done.**

*l88: Change "the salinity gradients in AS and BS seem to be more pronounced in observations" to "the observed salinity gradients in the AS and BS are more strongly pronounced than in the model results".*

**Done.**

*l92: Change "(Table 2; Depoorter et al. (2013); Rignot et al. (2013))." to "(Table 2; Depoorter et al., 2013x; Rignot et al., 2013)."*

**Due to Latex command, I was not able to make this change in bibtex. I think it is better to keep as it is, since it is possible to edit these points in postprocessing.**

*l94: Add "the" to read "on the 2006-2007 ice shelf configurations".*

**Done.**

*l95: Change "may represent better" to "may better represent".*

**Done.**

*l99: Add "the" to read "the melt flux decreases".*

**Done.**

*l99: Change "until the" to "towards the".*

**Done.**

*l100: Change "This temporal variability is similar to the PRS case." to "This evolves similarly to what was obtained in the PRS case."*

**Done.**

*l101: Change "does not show" to "does not exhibit".*

**Done.**

*l101: Change "remains similar at the value" to "remains at a value".*

**Done.**

*l106: Correct "the the RS" to "the RS".*

**Done.**

*l107: Change "(Fig. 5; Nakayama et al. (2014a); Dinniman et al. (2016))." to "(Fig. 5; Nakayama et al., 2014a; Dinniman et al., 2016)."*

**Due to Latex command, I was not able to make this change.**

*l108: Add info to where the glacial meltwater spreads: "further downstream of/to XXXX".*

**We think spreading patterns are already specified in the previous sentence.**

*l118: Change "warming and a" to "warming as well as a".*

**Done.**

*l119: Change "that we simulate" to "us simulating".*

**Done.**

*l122: Change "(see the black arrow in Fig. S3)" to "(Fig. S3, black arrow)".*

**Done.**

*l123: Change "occurs along with" to "occurs in parallel to".*

**Done.**

*l135: Change "deep RS" to "the deep RS".*

**Done.**

*l144: Change "roughly 5 more years" to "an additional five years".*

**Done.**

*l149: Change "Previous studies show" to "Previous studies showed".*

**Done.**

*l152: Change "is at least an order of magnitude greater than ice shelf melt over the observational period" to "would require an order of magnitude higher ice shelf melt rates for the duration of the observations".*

**Done.**

*l153: Change "We also note" to "Importantly we note".*

**Done.**

*l153: Change "spreading on sea ice" to "spread on the sea ice".*

**Done.**

*l154: Add "the" to read "the sea-ice extent".*

**Done.**

*l154: Hyphenation rules: This is the first time you use the hyphen: "sea-ice extent" ---> Need to be consistent throughout the ms.*

**Done.**

*l161: Change "We aim to investigate" to "We set out to investigate".*

**Done.**

*l176: Change "(e.g., Nakayama et al. (2019); Shean et al. (2019))" to "(e.g., Nakayama et al., 2019; Shean et al., 2019)".*

**Due to Latex command, I was not able to make this change.**

*l177: Change "and simulates on-shelf CDW intrusions possibly weaker than observations" to "and consequently simulated on-shelf CDW intrusions are weaker than those observed"*

**Done.**

*l180: This sentence would improve with some rewriting.*

**Done.**

*l188: Change "is high." to "peaks."????*

**We think it is better to keep "is high" here.**

*l189: Change "and a slow response of 15-20 years" to "and a relatively sluggish response over the 15 to 20 model years".*

**Done.**

*l189: Change "we are likely not able to extract the effect of enhanced ice shelf melting from the existing observations in these regions" to "the effect of enhanced ice shelf melting cannot clearly be detected in the existing observations for these regions".*

**Done.**

*l191: Add "observed" to read "of the observed salinity decrease".*

**Done.**

*l191: Change "today, circum-Antarctic freshening could be underway." to "by now, circum-Antarctic freshening is likely to occur".*

**Done.**

*l192: Remove "solely". It doubles up with "alone".*

**Done.**

*l193: Change "due to a strong density gradient" to "due to an increased density gradient".*

**Done.**

*l194: Change "(Nakayama et al. (2014a))" to "(Nakayama et al., 2014a)*

*References: I have not cross-checked these. Pls do so and correct as necessary.*

**Due to Latex command, I was not able to make this change.**

*Figures: They are very small. Can you provide larger figures?*

**Some figures are now enlarged.**

*Fig. 2: Add "L" to read "for the LMELT (L) case."*

**Done.**

*Fig. 2: Change "as black lines." to "in black."*

**Done.**

*Fig. 5: Change "as black lines." to "in black."*

**Done.**

*Fig. 6: Change "as black lines." to "in black."*

**Done.**

*Fig. 7: Change "as black lines." to "in black."*

**Done.**

*Fig. 9: Change "as black lines." to "in black."*

**Done.**

*Tab. 2: Should are increased for sensitivity experiments." read "are increased in the sensitivity experiments."?*

**Done.**

*Tab. 2: Separate references by semicolon not comma: "Rignot et al., 2013; Depoorter et al., 2013" 4a)".*

**Done.**

*l196: Change "However, for other cases," to "However for the other cases,".*

**Done.**

*l201: Change "converge towards the end of the simulation" to "towards the end of the simulation converge".*

**Done.**

*l203: Change "Further investigations with longer model integration" to "Further studies with longer model integration times". Ie. to avoid "investigations" and "investigate" in the one sentence.*

**Done.**

*l210: Change "can propagate further downstream" to "may propagate further downstream".*

**Since we add more information related to comment below, we did not make this modification.**

*l210: Further to what? --> "further in case x than in case y".*

**Done.**

*l214: "may impact": How? --> "may reduce the production of AABW".*

**Done.**

*l217: Change "when ice loss in the AS and BS is high." to "when ice loss in the AS and BS is significant."*

**Done.**

*l220: Change "for understanding" to "to understand".*

**Done.**

*l233: Remove "The model code, processing tools, and raw model output are difficult to make publicly available, and the authors recommend contacting the corresponding author for those interested in accessing the data."*

**Done.**

*l236: Correct "140 E" to "140 ^oE" (include degree sign).*

**Done.**

*References in general: Unsure if TC requires to include DOIs. --> Pls check.*

**I checked recently published paper and I think it is fine without DOIs.**

*References: I have not cross-checked these. Pls do so and correct as necessary.*

**Done.**

*Figures: They are very small. Can you provide larger figures?*

**Done.**

*Fig. 2: Add "L" to read "for the LMELT (L) case."*

**Done.**

*Fig. 2: Change "as black lines." to "in black."*

**Done.**

*Fig. 5: Change "as black lines." to "in black."*

**Done.**

*Fig. 6: Change "as black lines." to "in black."*

**Done.**

*Fig. 7: Change "as black lines." to "in black."*

**Done.**

*Fig. 9: Change "as black lines." to "in black."*

**Done.**

*Tab. 2: Should are increased for sensitivity experiments." read "are increased in the sensitivity experiments."?*

**Done.**

*Tab. 2: Separate references by semicolon not comma: "Rignot et al., 2013; Depoorter et al., 2013".*

**Done.**

[revised manuscript text omitted]